# A quantum processor based on coherent transport of entangled atom arrays

Dolev Bluvstein[1], Harry Levine[1,6], Giulia Semeghini[1], Tout T. Wang[1], Sepehr Ebadi[1], Marcin Kalinowski[1], Alexander Keesling[1,2], Nishad Maskara[1], Hannes Pichler[3,4], Markus Greiner[1], Vladan Vuletić[5] & Mikhail D. Lukin[1✉]

The ability to engineer parallel, programmable operations between desired qubits within a quantum processor is key for building scalable quantum information systems[1,2]. In most state-of-the-art approaches, qubits interact locally, constrained by the connectivity associated with their fixed spatial layout. Here we demonstrate a quantum processor with dynamic, non-local connectivity, in which entangled qubits are coherently transported in a highly parallel manner across two spatial dimensions, between layers of single- and two-qubit operations. Our approach makes use of neutral atom arrays trapped and transported by optical tweezers; hyperfine states are used for robust quantum information storage, and excitation into Rydberg states is used for entanglement generation[3–5]. We use this architecture to realize programmable generation of entangled graph states, such as cluster states and a seven-qubit Steane code state[6,7]. Furthermore, we shuttle entangled ancilla arrays to realize a surface code state with thirteen data and six ancillary qubits[8] and a toric code state on a torus with sixteen data and eight ancillary qubits[9]. Finally, we use this architecture to realize a hybrid analogue–digital evolution[2] and use it for measuring entanglement entropy in quantum simulations[10–12], experimentally observing non-monotonic entanglement dynamics associated with quantum many-body scars[13,14]. Realizing a long-standing goal, these results provide a route towards scalable quantum processing and enable applications ranging from simulation to metrology.

Quantum information systems derive their power from controllable interactions that generate quantum entanglement. However, the natural, local character of interactions limits the connectivity of quantum circuits and simulations. Non-local connectivity can be engineered via a global shared quantum data bus[15–18], but in practice these approaches have been limited in either control or size. A number of visionary architectures to address this challenge have been proposed theoretically over the past two decades. On the basis of coherent, dynamical transport of quantum information using movable traps or photonic links, these techniques have been the subject of intensive experimental explorations across different platforms[1,19–24]. However, progress has been limited to small-scale, few-qubit systems lacking either full connectivity, programmability or true parallelism.

Our approach to address this long-standing challenge utilizes dynamically reconfigurable arrays of entangled neutral atoms, shuttled by optical tweezers in two spatial dimensions (Fig. 1a). Hyperfine states are used for storing and transporting quantum information between quantum operations, and excitation into Rydberg states is used for generating entanglement. Highly parallel operations are enabled via selective qubit operations in distinct zones that qubits are dynamically shuttled between. Taken together, these ingredients enable a

powerful quantum information architecture, which we employ to realize applications including entangled-state generation, the creation of topological surface and toric code states, and hybrid analogue–digital quantum simulations.

## Entanglement transport in atom arrays

Our experiments utilize a two-dimensional (2D) atom array system described previously[25], with key upgrades to enable coherent transport and multiple layers of single-qubit and two-qubit gates. We store quantum information in magnetically insensitive clock states within the ground-state hyperfine manifold of [87]Rb atoms[20], and implement robust single-qubit Raman rotations (scattering error per π pulse of about $7 \times 10^{-5}$)[26], realized by composite pulses that are robust to pulse errors (Extended Data Fig. 3)[27]. High-fidelity controlled-Z (CZ) entangling gates in the hyperfine basis $\{|0\rangle, |1\rangle\}$ (Fig. 1a) are implemented in parallel using global Rydberg excitation pulses on the $|1\rangle \leftrightarrow |r\rangle$ Rydberg transition[5]. For dynamic reconfiguration, we initialize atoms into two sets of traps: static traps generated by a spatial light modulator (SLM) and mobile traps generated by a crossed 2D acousto-optic deflector (AOD). To execute a specific circuit, we arrange qubits into desired pairs, perform Rydberg-mediated CZ gates on

[1]Department of Physics, Harvard University, Cambridge, MA, USA. [2]QuEra Computing Inc., Boston, MA, USA. [3]Institute for Theoretical Physics, University of Innsbruck, Innsbruck, Austria. [4]Institute for Quantum Optics and Quantum Information, Austrian Academy of Sciences, Innsbruck, Austria. [5]Department of Physics and Research Laboratory of Electronics, Massachusetts Institute of Technology, Cambridge, MA, USA. [6]Present address: AWS Center for Quantum Computing, Pasadena, CA, USA. ✉e-mail: lukin@physics.harvard.edu

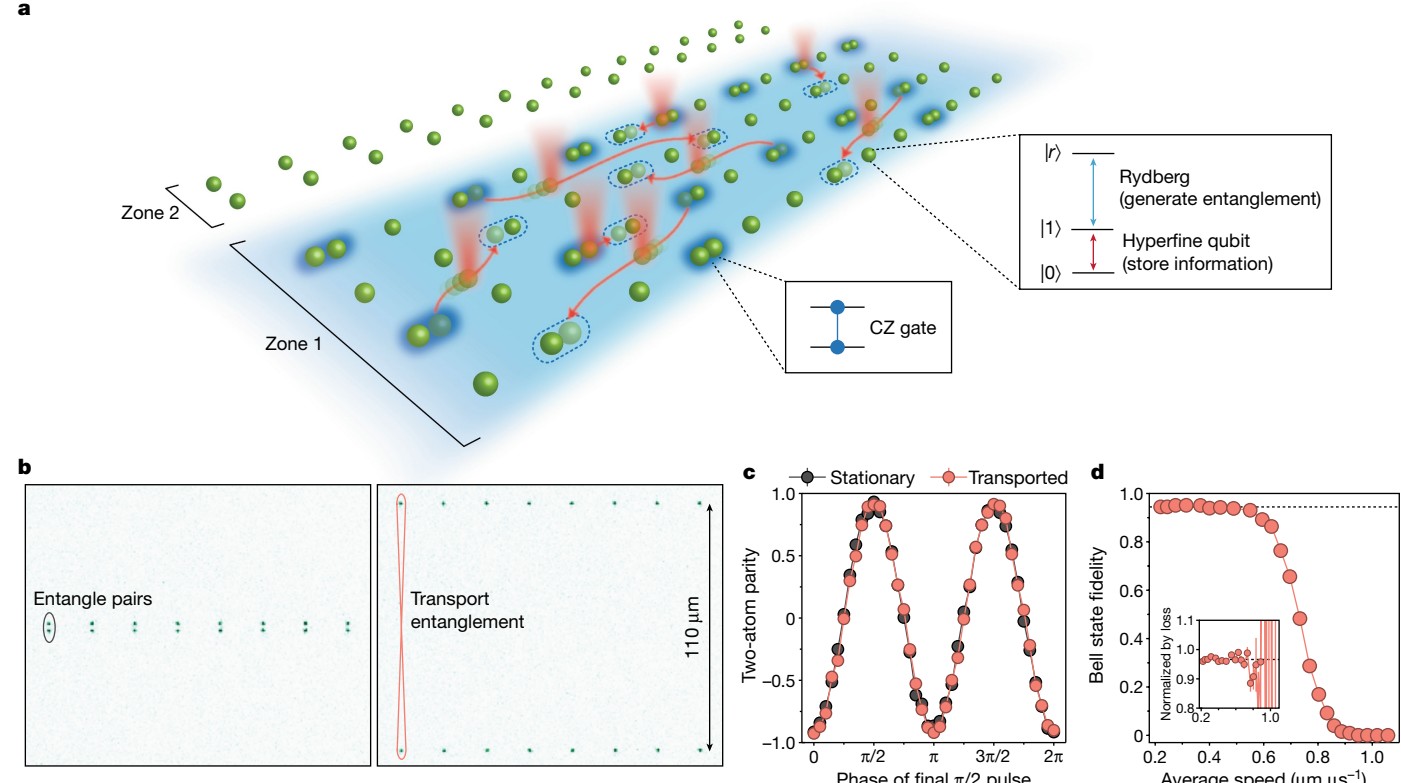

**Fig. 1 | Quantum information architecture enabled by coherent transport of neutral atoms. a**, In our approach, qubits are transported to perform entangling gates with distant qubits, enabling programmable and non-local connectivity. Atom shuttling is performed using optical tweezers, with high parallelism in two dimensions and between multiple zones allowing selective manipulations. Inset: the atomic levels used. The $|0\rangle$, $|1\rangle$ qubit states refer to the $m_F = 0$ clock states of $^{87}$Rb, and $|r\rangle$ is a Rydberg state used for generating entanglement between qubits (Extended Data Fig. 1b). **b**, Atom images illustrating coherent transport of entangled qubits. Using a sequence of single-qubit and two-qubit gates, atom pairs are each prepared in the $|\Phi^+\rangle$ Bell

state (Methods), and are then separated by 110 μm over a span of 300 μs. **c**, Parity oscillations indicate that movement does not observably affect entanglement or coherence. For both the moving and the stationary measurements, qubit coherence is preserved using an XY8 dynamical decoupling sequence for 300 μs (Methods). **d**, Measured Bell-state fidelity as a function of separation speed over the 110 μm, showing that fidelity is unaffected for a move slower than 200 μs (average separation speed of 0.55 μm μs$^{-1}$). Inset: normalizing by atom loss during the move results in constant fidelity, indicating that atom loss is the dominant error mechanism (see Methods for details).

each pair simultaneously and then move all mobile traps in parallel to dynamically change the connectivity into the next desired qubit arrangement.

Figure 1 shows our ability to transport qubits across large distances while preserving entanglement and coherence[20]. We initialize pairs at an atom–atom distance of 3 μm (Fig. 1b) and then create a Bell state $|\Phi^+\rangle = \frac{1}{\sqrt{2}}(|00\rangle + |11\rangle)$ in the hyperfine basis (Methods)[5]. To measure the resulting entangled-state fidelity, we apply a variable single-qubit phase gate before a final π/2 pulse, resulting in oscillations of the two-atom parity $\langle \sigma_1^z \sigma_2^z \rangle$ (Fig. 1c)[5]. We then repeat this experiment, but now move the atoms apart by 110 μm before applying the final π/2 pulse. Our transport protocol is optimized to suppress heating and loss by implementing cubic-interpolated atom trajectories (Methods), and is further accompanied by an eight-pulse XY8 robust dynamical decoupling sequence[28] to suppress dephasing. The resulting parity oscillations indicate that two-atom entanglement is unaffected by the transport process[20,29]. Performing this experiment as a function of movement speed[30] shows that the fidelity remains unchanged until the total separation speed becomes more than 0.55 μm μs$^{-1}$, corresponding to the onset of atom loss (Fig. 1d). We note that the entanglement transport in Fig. 1b corresponds to moving quantum information across a region of space that can, in principle, host about 2,000 qubits (at an atom separation of 3 μm), on a timescale corresponding to <10$^{-3}$ of the coherence time $T_2$ (Extended Data Fig. 3), directly enabling applications in large-scale quantum information systems.

## Programmable circuits and graph states

To exemplify the ability to generate non-local connectivity between qubit arrays in parallel, we carry out the preparation of entangled graph states: a large class of useful quantum information states, with examples ranging from Greenberger–Horne–Zeilinger states and cluster states to quantum error correction (QEC) codes[31]. Graph states are defined by initializing all qubits, located on the vertices of a geometric graph, in $|+\rangle = \frac{|0\rangle + |1\rangle}{\sqrt{2}}$ and then performing CZ gates on the links between qubits (corresponding to the edges of the graph)[31]. $N$-qubit graph states $|G\rangle$ are associated with a set of $N$ stabilizers, defined by $S_i = X_i \Pi_{j \in u_i} Z_j$, where X and Z are the Pauli matrices, $u_i$ is the set of qubits (vertices) connected by an edge to qubit $i$, and $\Pi$ denotes a product over qubit indices $j$ (ref. [31]). The stabilizers each have +1 eigenvalue for the graph state $|G\rangle$. Measuring these operators and their expectation values can be used to characterize the preparation of the target state.

As an example, Fig. 2a shows the preparation of a one-dimensional (1D) cluster state, a graph state defined by a linear chain of qubits[32,33]. To realize this state, we perform one global, parallel layer of CZ gates on adjacent atom pairs, move half the atoms to form new pairs and then perform another parallel layer of CZ gates (Fig. 2a, b). To probe the resultant 12-qubit cluster state, we measure the stabilizer set $\{S_i\} = \{Z_{i-1} X_i Z_{i+1}\}$ through readout in two measurement settings, given by a local π/2 rotation on either the odd or the even sublattice before projective measurement[34]. We achieve the local rotation by moving one sublattice of qubits to a separate zone and

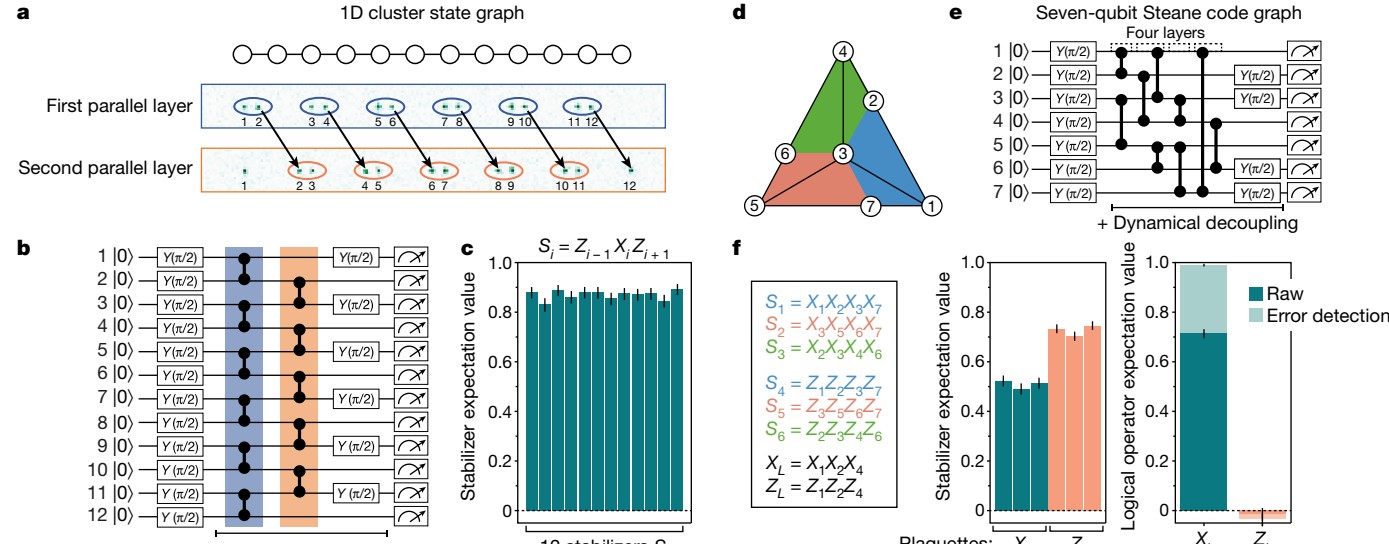

**Fig. 2 | One- and two-dimensional graph states using dynamic entanglement transport. a**, Generation of a 12-atom 1D cluster-state graph, created by initializing all qubits (vertices) in $|+\rangle$ and applying CZ gates on the links (edges) between qubits. The atom images show the configuration for the first and second gate layers. **b**, Quantum circuit representation of the 1D cluster-state preparation and measurement. Dynamical decoupling is applied throughout all quantum circuits (Methods). **c**, Raw measured stabilizers of the resulting 1D cluster state, given by $S_i = Z_{i-1}X_iZ_{i+1}$ ($X_1Z_2$ and $Z_{11}X_{12}$ for the edge

qubits). **d**, Graph-state representation of the seven-qubit Steane code (colours represent stabilizer plaquettes). **e**, Circuit for preparing the Steane code logical $|+\rangle_L$ state, performed in four parallel gate layers. **f**, Measured stabilizers and logical operators after preparing $|+\rangle_L$. Error detection is done by postselecting on measurements where all stabilizers are +1. For both the 1D cluster state and the Steane code, the stabilizers and logical operators are measured with two measurement settings (see text). Error bars represent 68% confidence intervals.

then performing a rotation on the unmoved qubits with a homogeneous beam illuminating the experiment zone (Fig. 1a, Methods). We measure $\langle S_i\rangle$ by analysing the resulting bit-string outputs and plot the resulting raw stabilizer measurements (Fig. 2c). Across all 12 stabilizers, we find an average $\langle S_i\rangle = 0.87(1)$ (Fig. 2c) (accounting for state-preparation-and-measurement (SPAM) errors would yield $\langle S_i\rangle = 0.91(1)$), certifying biseparable entanglement in a cluster state (all $\langle S_i\rangle > 0.5$ (ref. [34])). The measured fidelities would correspond to a few-percent error per operation for a measurement-based quantum computation[32,35].

An important class of graph states are QEC codes, where the graph-state stabilizers manifest as the stabilizers of the QEC code and can be measured to correct errors on an encoded logical qubit. As an example, we prepare the seven-qubit Steane code[6,7], a topological colour code depicted by the graph in Fig. 2d, in the logical state $|+\rangle_L$. To prepare this state, we initialize all qubits in $|+\rangle$, apply CZs on the links between qubits (in four parallel layers; Extended Data Fig. 5) and then rotate either of the two sublattices for measuring stabilizers (Fig. 2e). After sublattice rotation, six of the graph-state stabilizers transform into the six Steane code stabilizers, given by four-body products of $X_i$ or $Z_i$. Figure 2f shows the raw measured expectation values of these six stabilizers. The seventh graph-state stabilizer transforms into the logical qubit operator $X_L$ and has eigenvalue +1 for the graph state $|G\rangle$, while anticommuting with logical $Z_L$. Accordingly, in Fig. 2f, we find $\langle X_L\rangle = 0.71(2)$ and $\langle Z_L\rangle = -0.02(3)$, demonstrating the preparation of the logical qubit state $|+\rangle_L$. Moreover, we perform error detection by post-selecting on measurement outcomes where all measured stabilizers yield +1 (refs. [36,37]; with 66(1)% probability of no detected errors). Using this procedure, we obtain corrected values of $\langle \overline{X_L}\rangle = 0.991^{+0.004}_{-0.007}$ and, $\langle \overline{Z_L}\rangle = -0.03(3)$ demonstrating the error-detecting properties of the Steane code graph (see Extended Data Fig. 7 for error correction and logical operations).

## Topological states with ancilla arrays

We next make use of transportable ancillary qubit arrays to mediate quantum operations between remote qubits[1]. Owing to the ability to

quickly move arrays of atoms across the entire system, the use of ancillary qubits naturally complements our movement capabilities. Specifically, we employ ancillas for state preparation by mediating entanglement between physical qubits that never directly interact, followed by projective measurement of the ancilla array (performed simultaneously with the measurement of the data qubits), a form of measurement-based quantum computation[32,35]. In particular, we prepare topological surface code and toric code states[8,9,38,39], whose states are more difficult to construct by direct CZ gates between physical qubits (requiring an extensive number of layers[8,40]). For these codes, the measured values of the ancilla qubits simply redefine the stabilizers and are handled in-software for practical QEC operation[38]. As the redefinition is applied in-software, without physical intervention, the projective measurements on the ancillae commute with all operations on the data qubits and can be done at any time, and so we measure all qubits simultaneously.

Figure 3a shows the preparation of a 19-qubit graph state creating the $|+\rangle_L$ logical state of the surface code[8,38]. The surface code is defined by $X$-plaquette and $Z$-star stabilizers, and logical operators $X_L$ ($Z_L$) are defined as strings of $X$ ($Z$) products across the height (width) of the graph. To prepare this state, ancillas are moved to perform CZ gates with each of their four neighbours and are then measured, projecting the data qubits into the surface code state. The graph-state stabilizers now transform into the $X$ plaquettes, the $Z$ stars (with value ±1 for a measurement outcome of ±1 of the central ancilla) and the logical $X_L$ operator[35,41]. Remarkably, this procedure creates a topologically ordered state in a constant-depth circuit[35,40], where measured ancilla values can be used for redefining stabilizers, which can be handled in-software for practical QEC operation[38]. Figure 3b shows the measured expectation values of the 12 resulting stabilizers, as well as the logical operator expectation values with and without error detection. We find a raw value of $\langle X_L\rangle = 0.64(3)$, and a corrected value of $\langle \overline{X_L}\rangle = 1^{+0}_{-0.01}$ using the measured stabilizers for error detection (with 35(1)% probability of no detected errors), demonstrating the preparation of this topological QEC state (see also Extended Data Fig. 7, showing the expected attributes for all prepared error-protected logical states).

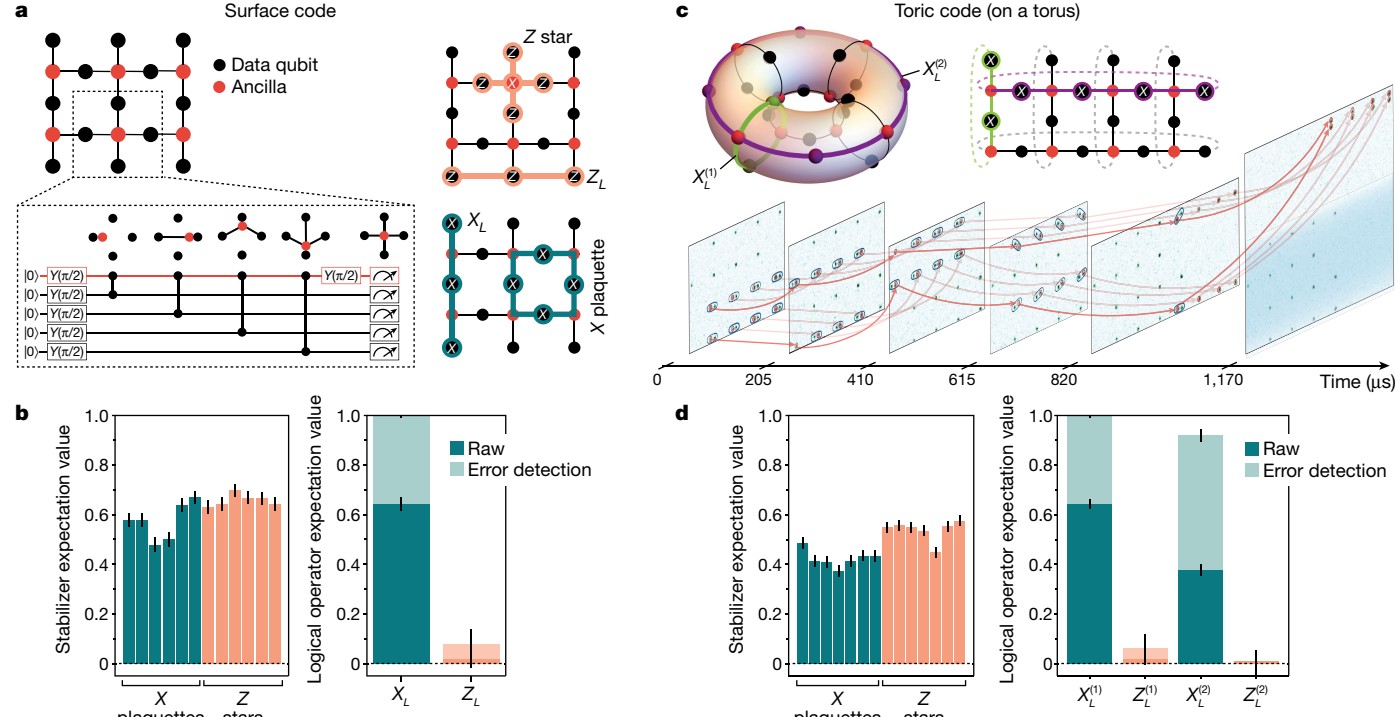

**Fig. 3 | Topological surface code and toric code states using mobile ancilla qubit arrays. a**, Graph state realizing the surface code. Left: the circuit depicts formation of the graph state by use of mobile ancilla qubits; each move corresponds to performing a CZ gate with a neighbouring data qubit (illustrated in box). The logical $|+\rangle_L$ state is created upon projective measurement of the ancilla qubits in the $X$ basis. Right: stabilizers and logical operators of the code. **b**, Measured $X$-plaquette and $Z$-star stabilizers of the resultant surface code, along with logical operators with and without error detection (implemented in postselection). **c**, Implementation of the toric code.

Top: graph state realizing the two logical-qubit product state $|+\rangle_L^{(1)}|+\rangle_L^{(2)}$ of the toric code upon projective measurement of the ancilla qubits in the $X$ basis. Bottom: images showing the movement steps implemented in creating and measuring the toric code state (Supplementary Video 1). The blue shading in the final image represents a local rotation on the data qubit zone. **d**, Measured $X$-plaquette and $Z$-star stabilizers, along with logical operators for the two logical qubits with and without error detection (implemented in postselection).

Although surface code states have previously been prepared with other methods, our transport capabilities allow us to use the full range of motion of ancilla qubits across the entire qubit array to enable periodic boundary conditions and realize the toric code state on a torus[9]. To this end, we create the 24-qubit graph state shown in Fig. 3c by performing five layers of parallel gates and moving the ancillae to their separate zone for readout in a separate basis (see also Supplementary Video 1 showing the full atom trajectory). The state we prepare has seven (owing to periodic boundary conditions) independent $X$ plaquettes and seven independent $Z$ stars. Moreover, owing to the topological properties of this graph, two independent logical qubits can be encoded with logical operators $X_L^{(1)}$, $Z_L^{(1)}$ and $X_L^{(2)}$, $Z_L^{(2)}$ that wrap around the entire torus along two topologically distinct directions[9]. Upon projective measurement of the ancilla qubits in the $X$ basis we create the toric code state $|+\rangle_L^{(1)}|+\rangle_L^{(2)}$. State preparation is verified in Fig. 3d by measuring the toric code stabilizers. For the two encoded logical qubits, we find raw logical qubit expectation values of $\langle X_L^{(1)}\rangle = 0.64(2)$ and $\langle X_L^{(2)}\rangle = 0.38(2)$, and error-detected values of $\langle \bar{X}_L^{(1)}\rangle = 1_{-0.01}^{+0}$ and $\langle \bar{X}_L^{(2)}\rangle = 0.92_{-0.03}^{+0.02}$ (with 20(1)% probability of no detected errors), demonstrating the preparation of the toric code. We note that the different expectation values of the corrected logical qubits originate from the aspect ratio of our torus, where $X_L^{(1)}$ and $X_L^{(2)}$ are protected to code distance $d = 4$ and $d = 2$, respectively (see also Extended Data Fig. 7). Our measured fidelities are in good agreement with numerical simulations of the circuit (Extended Data Fig. 6), wherein each qubit experiences a per-layer error rate independent of the number of qubits or the shuttling process, indicating that errors in CZ gates (fidelity of about 97.5%; Methods[5]) constitute our dominant error source.

## Hybrid analogue–digital circuits

Having established atom movement for realizing digital circuits, we now explore the applications to quantum simulation. In particular, we perform hybrid, modular quantum circuits composed of analogue Hamiltonian evolution, reconfiguration and digital gates (Fig. 4a). Together, these tools open a wide variety of possibilities in quantum simulation and many-body physics. As a specific example, we measure the Renyi entanglement entropy after a quantum quench by effectively interfering two copies of a many-body system[10,11].

Figure 4b illustrates the experimental procedure. After initializing both copies with all qubits in $|1\rangle$, we independently evolve each copy under the Rydberg Hamiltonian $H_{Ryd}$ for a time $t$, generating an entangled many-body state in the $\{|1\rangle, |r\rangle\}$ basis (Methods)[13]. Raman and Rydberg π pulses then map $|1\rangle \rightarrow |0\rangle$ and $|r\rangle \rightarrow |1\rangle$, transferring the entangled many-body state into the long-lived and non-interacting $\{|0\rangle, |1\rangle\}$ basis[42]. Finally, we measure entanglement entropy by rearranging the system and interfering each qubit in the first copy with its identical twin in the second copy, by use of a Bell measurement circuit. Measuring twins in the Bell basis detects occurences of the antisymmetric singlet state $|\Psi^-\rangle = \frac{|01\rangle - |10\rangle}{\sqrt{2}}$, the presence of which indicates that subsystems of the two copies were in different states owing to entanglement with the rest of the many-body system[10,11]. Quantitatively, analysing the number parity of observed singlets within subsystem A yields the purity $\mathrm{Tr}\left[\rho_A^2\right]$ of the reduced density matrix $\rho_A$, and thus yields the second-order Renyi entanglement entropy $S_2(A) = -\log_2 \mathrm{Tr}\left[\rho_A^2\right]$ (Methods). This measurement circuit provides the Renyi entropy of any constituent subsystem of our whole closed quantum system, where

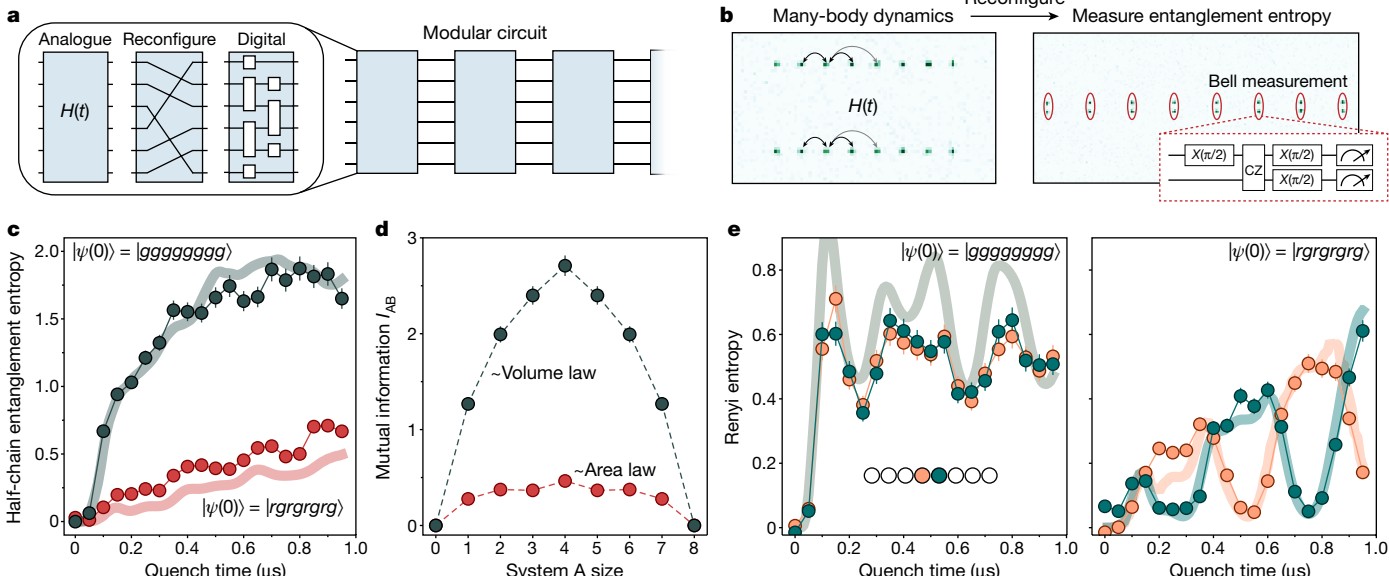

**Fig. 4 | Dynamic reconfigurability for hybrid analogue–digital quantum simulation. a**, Hybrid quantum circuit combining coherent atom transport with analogue Hamiltonian evolution and digital quantum gates. **b**, Measuring entanglement entropy in a many-body Rydberg system via two-copy interferometry. **c**, Measured half-chain Renyi entanglement entropy after many-body dynamics following quenches on two eight-atom systems. Quenching from $|gggg...\rangle$ ($|g\rangle \equiv |1\rangle$) results in rapid entropy growth and saturation, signifying quantum thermalization. Quenching from $|rgrg...\rangle$ reveals a significantly slower growth of entanglement entropy. **d**, Measuring

the mutual information at 0.5-μs quench time reveals a volume-law scaling for the thermalizing $|gggg...\rangle$ state, and an area-law scaling for the scarring $|rgrg...\rangle$ state. **e**, The single-site Renyi entropies for sites in the middle of the chain quickly increase and saturate for the $|gggg...\rangle$ quench, but show large oscillations for the $|rgrg...\rangle$ quench. The solid curves are results of exact numerical simulations for the isolated quantum system under $H_{\text{Ryd}}$ with no free parameters (see Methods for details of data processing). Error bars represent one standard deviation.

the calculation over any desired subsystem A is simply performed in data processing[10,11].

We use this method to probe the growth of entanglement entropy produced by many-body dynamics (see Methods for additional benchmarking of the technique). Specifically, we study the evolution of two eight-atom copies under the Rydberg Hamiltonian, subject to the nearest-neighbour blockade constraint[4,13]. Upon a rapid quench from an initial state with all atoms in the ground state $|g\rangle \equiv |1\rangle$, we observe that the half-chain Renyi entanglement entropy quickly grows and saturates (Fig. 4c), a process corresponding to quantum thermalization[12]. By analysing the Renyi mutual information $I_{AB} = S_2(A) + S_2(B) - S_2(AB)$ between the leftmost $n$ atoms in the chain (A) and the complement subsystem of the rightmost $8 - n$ atoms (B), we find a volume-law scaling in the resulting state (Fig. 4d)[11,12].

Although such thermalizing dynamics is generically expected in strongly interacting many-body systems, remarkably, it was demonstrated previously that for certain initial states this system can evade thermalization. Underpinned by special, non-thermal eigenstates called quantum many-body scars[13,14,43], these states were theoretically predicted to feature dynamics associated with a slow, non-monotonic entanglement growth. Figure 4 shows the measurement of entanglement properties of many-body scars following a rapid quench from the initial state $|Z_2\rangle \equiv |rgrg...\rangle$, initialized by applying local Rydberg π pulses (Methods). We find that the rate of entropy growth for this initial state is significantly suppressed, and the mutual information reveals an area-law scaling (Fig. 4d). Furthermore, Fig. 4e shows the single-site entropy in the middle of the chain, demonstrating rapid growth and saturation for the thermalizing $|gggg...\rangle$ state but large oscillations for the $|Z_2\rangle$ state[13,14]. Remarkably, the data show that when sites of one sublattice return to low entropy, the other sublattice goes to high entropy; this reveals that the scar dynamics entangle distant atoms (of the same sublattice) while disentangling nearest neighbours, even with only nearest-neighbour interactions (Methods). These measurements reveal non-trivial aspects

of quantum many-body scars, and constitute the direct observation of exotic entanglement phenomena in a many-body system.

These observations are in excellent agreement with exact numerical simulations in the isolated system (lines plotted in Fig. 4c, e, Extended Data Fig. 10). Moreover, whereas the single-site purity approaches that of a fully mixed state, our global purity (a 16-body observable composed of three-level systems) remains more than 100 times that of a fully mixed state (Extended Data Fig. 9), altogether demonstrating the high accuracy and fidelity of our circuit-based technique. These results demonstrate that combining atom movement, many-body Hamiltonian evolution and digital quantum circuits yields powerful tools for simulating and probing the quantum physics of complex systems.

## Discussion and outlook

Our experiments demonstrate highly parallel coherent qubit transport and entanglement enabling a powerful quantum information architecture. The present techniques can be extended along a number of directions. Local Rydberg excitation on subsets of qubit pairs would eliminate residual interactions from unintended atoms, allowing parallel, independent operations on arrays with significantly higher qubit densities. Two-qubit gate fidelity can be improved using higher Rydberg laser power or more efficient delivery methods, as well as more advanced atom cooling[44]. These technical improvements should allow for scaling to deep quantum circuits operating on thousands of neutral atom qubits. These upgrades can be additionally supplemented by more sophisticated local single-qubit control employing, for example, parallel Raman excitation through acousto-optic modulator arrays[16]. Mid-circuit readout can be implemented by moving ancillas into a separate zone and imaging using, for example, avalanche photodiode arrays within a few hundred microseconds[45].

Our method has a clear potential for realizing scalable QEC[46]. For example, the procedure demonstrated in Fig. 3c can be used for

syndrome extraction in a practical QEC sequence, wherein ancillas are entangled with their data qubit neighbours and then moved to a separate zone for mid-circuit readout. We estimate that an entire QEC round can be implemented within a millisecond, much faster than the measured $T_2 > 1$ s, and with projected fidelity improvements theoretically surpassing the surface code threshold (Methods). We emphasize that such a mid-circuit readout is essential for realizing scalable fault-tolerant quantum computation. Furthermore, the ability to reconfigure and interlace our arrays will allow efficient, parallel execution of transversal entangling gates between many logical qubits[38,47]. In addition, these techniques also enable implementation of higher-dimensional or non-local error-correcting codes with more favourable properties[48,49]. Together, these ingredients could enable an approach to universal, fault-tolerant quantum computing with thousands of physical qubits.

Our dynamically reconfigurable architecture also opens many opportunities for digital and analogue quantum simulations. For example, our hybrid approach can be extended to probing the entire entanglement spectrum[50], simulating wormhole creation[51], performing many-body purification[52] and engineering non-equilibrium states[53]. Entanglement transport could also empower metrological applications such as creating distributed states for probing gravitational gradients[54]. Finally, our approach can facilitate quantum networking between separated arrays, paving the way towards large-scale quantum information systems[29,55] and distributed quantum metrology[54,56].

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

# Methods

## Dynamic reconfiguration in 2D tweezer arrays

Our experiments utilize the same apparatus described previously in ref. [25]. Inside our vacuum cell, $^{87}$Rb atoms are loaded from a magneto-optical trap into a backbone array of programmable optical tweezers generated by an SLM[57]. Atoms are rearranged in parallel into defect-free target positions in this SLM backbone[57] by additional optical tweezers generated from a crossed 2D AOD. Following the rearrangement procedure, we transfer selected atoms from the static SLM traps back into the mobile AOD traps, and then move these mobile atoms to their starting positions in the quantum circuit. During this entire process, the atoms are cooled with polarization-gradient cooling. Before running the quantum circuit, we take a camera image of the atoms in their initial starting positions, and following the circuit we take a final camera image to detect qubit states |0⟩ (atom presence) and |1⟩ (atom loss, following resonant pushout). We postselect all data on finding perfect rearrangement of the AOD and SLM atoms before running the circuit. In all experiments here, each atom remains in a single static or single mobile trap throughout the duration of the quantum circuit[20,58,59].

The crossed AOD system is composed of two independently controlled AODs (AA Opto Electronic DTSX-400) for $x$ and $y$ control of the beam positions. Both AODs are driven by independent arbitrary waveforms, which are generated by a dual-channel arbitrary waveform generator (M4i.6631-x8 by Spectrum Instrumentation) and then amplified through independent MW amplifiers (Minicircuits ZHL-5W-1). The time-domain arbitrary waveforms are composed of multiple frequency tones corresponding to the $x$ and $y$ positions of columns and rows, which are independently changed as a function of time for steering around the AOD-trapped atoms dynamically; the full $x$ and $y$ waveforms are calculated by adding together the time-domain profile of all frequency components with a given amplitude and phase for each component. For running quantum circuits, we program the positions of the AOD atoms at each gate location and then smoothly interpolate (with a cubic profile) the AOD frequencies as a function of time between gate positions. The cubic profile enacts a constant jerk onto the atoms, which allows us to move roughly five-to-ten-times faster (without heating and loss) than if we move at a constant velocity (linear profile). In our movement protocol, we only do stretches, compressions and translations of the AOD trap array: that is, the AOD rows and columns never cross each other to avoid atom loss and heating associated with two frequency components crossing each other.

We homogenize the AOD tweezer intensity throughout the whole atom trajectory to minimize dephasing induced by a time-varying magnitude of differential light shifts. To this end, we use a reference camera in the image plane to gauge the intensity of each AOD tweezer at each gate location and homogenize by varying the amplitude of each frequency component; during motion between two locations, we interpolate the amplitude of each individual frequency component.

The SLM tweezer light (830 nm) and the AOD tweezer light (828 nm) are generated by two separate, free-running titanium:sapphire lasers (M Squared, 18-W pump). Projected through a 0.5 numerical aperture objective, the SLM tweezers have a waist of roughly 900 nm (roughly 1,000 nm for AODs). When loading the atoms, the trap depths are about $2\pi \times 16$ MHz, with radial trap frequencies of about $2\pi \times 80$ kHz, and when running quantum circuits the trap depths are about $2\pi \times 4$ MHz, with radial trap frequencies of about $2\pi \times 40$ kHz.

## Raman laser system

Fast, high-fidelity single-qubit manipulations are critical ingredients of the quantum circuits demonstrated in this work. To this end, we use a high-power 795-nm Raman laser system for driving global single-qubit rotations between magnetic sublevel $m_F = 0$ clock states. This Raman laser system is based on dispersive optics, developed and described in ref. [26]. The 795-nm light (Toptica TA pro, 1.8 W) is phase-modulated by an electro-optic modulator (Qubig), which is driven by microwaves at 3.4 GHz (Stanford Research Systems SRS SG384) that are doubled to 6.8 GHz and amplified. The laser phase modulation is converted to amplitude modulation for driving Raman transitions through use of a chirped Bragg grating (Optigrate)[26]. IQ (in-phase and quadrature) control of the SG384 is used for frequency and phase control of the microwaves, which are imprinted onto the laser amplitude modulation and thus give us direct frequency and phase control over the hyperfine qubit drive.

The Raman laser illuminates the atom plane from the side in a circularly polarized elliptical beam with waists of 40 μm and 560 μm on the thin axis and the tall axis, respectively, with a total average optical power of 150 mW on the atoms. The large vertical extent ensures <1% inhomogeneity across the atoms, and shot-to-shot fluctuations in the laser intensity are also <1%. For Figs. 1–3, we operate our Raman laser at a blue-detuned intermediate-state detuning of 180 GHz, resulting in two-photon Rabi frequencies of 1 MHz and an estimated scattering error per π pulse of $7 \times 10^{-5}$ (that is, 1 scattering event per 15,000 π pulses)[26]. For Fig. 4, to shorten the duration of the coherent mapping pulse sequence, we increase the Raman laser power and operate at a smaller blue-detuned intermediate-state detuning of 63 GHz, with a corresponding two-photon Rabi frequency of 3.2 MHz and an estimated scattering error per π pulse of $2 \times 10^{-4}$.

## Robust single-qubit rotations

For almost all single-qubit rotations in this work (other than XY8 and XY16 self-correcting sequences), we implement robust single-qubit rotations in the form of composite pulse sequences. These composite pulse sequences are well known in the NMR community[27,60] and can be highly insensitive to pulse errors such as amplitude or detuning miscalibrations. Our dominant source of coherent single-qubit errors arise from ≲1% amplitude drifts and inhomogeneity across the array; as such, we primarily use the 'BB1' (broadband 1) pulse sequence, which is a sequence of four pulses that implements an arbitrary rotation on the Bloch sphere while being insensitive to amplitude errors to sixth order[27,60]. We benchmark the performance of these robust pulses in Extended Data Fig. 3a. Furthermore, by applying a train of BB1 pulses, we find an accumulated error consistent with the estimated scattering limit (not plotted here), suggesting that the scattering limit roughly represents our single-qubit rotation infidelities (about $3 \times 10^{-4}$ error per BB1 pulse owing to the increased length of the composite pulse sequence). Randomized benchmarking[61] can be applied in future studies to further study single-qubit rotation fidelity.

## Qubit coherence and dynamical decoupling

In our 830-nm traps, hyperfine qubit coherence is characterized by inhomogeneous dephasing time $T_2^* = 4$ ms (not plotted here), $T_2 = 1.5$ s (XY16 with 128 total π pulses) and relaxation time $T_1 = 4$ s (including atom loss) (Extended Data Fig. 3b, c). All of our experiments in this work are performed in a d.c. magnetic field of 8.5 G. Coherence can be further improved by using further-detuned optical tweezers (with trap depth held constant, the tweezer differential lightshifts decrease as $1/\Delta$ and $1/T_1$ decreases as $1/\Delta^3$ (ref. [62]), where $\Delta$ is the detuning of the trap wavelength) and shielding against magnetic field fluctuations. For practical QEC operation, atom loss can be detected in a hardware-efficient manner[46] and the atom then replaced from a reservoir, which could in principle be continuously reloaded by a magneto-optical trap for reaching arbitrarily deep circuits.

All of our transport sequences[20,58,59] are accompanied by dynamical decoupling sequences. The number of pulses we use is a trade-off between preserving qubit coherence while minimizing pulse errors. We interchange between two types of dynamical decoupling sequence: XY8 and XY16 sequences, composed of phase-alternated individual π pulses that are self-correcting for amplitude and detuning errors[28,63], and Carr–Purcell–Meiboom–Gill (CPMG)-type dynamical decoupling

sequences composed of robust BB1 pulses. The CPMG-BB1 sequence is more robust to amplitude errors but incurs more scattering error. We empirically optimize for any given experiment by choosing between these different sequences and a variable number of decoupling π pulses, optimizing on either single-qubit coherence (including the movement) or the final signal. Typically, our decoupling sequences are composed of a total 12–18 π pulses.

## Movement effects on atom heating and loss

We study here the effects of movement on atom loss and heating in the harmonic oscillator potential given by the tweezer trap. Motion of the trap potential is equivalent to the non-inertial frame of reference where the harmonic oscillator potential is stationary but the atom experiences a fictitious force given by $F(t) = -ma(t)$, where $m$ is the mass of the particle and $a(t)$ is the acceleration of the trap as a function of time[64,65]. By following ref. [66] (equation 5.4), we find the average vibrational quantum number increase $\Delta N$ is given by

$$\Delta N = \frac{|\tilde{a}(\omega_0)|^2}{(2x_{\text{zpf}}\omega_0)^2}, \tag{1}$$

where $\tilde{a}(\omega_0)$ is the Fourier transform of $a(t)$ evaluated at the trap frequency $\omega_0$, and the zero-point size of the particle $x_{\text{zpf}} \equiv \sqrt{\hbar/(2m\omega_0)}$, where $\hbar$ is the reduced Planck constant. $\Delta N$ is the same for all initial levels of the oscillator[66]. Experimentally, we apply an acceleration profile $a(t) = jt$ to the atom, from time $-T/2$ to $+T/2$ to move a distance $D$ with constant jerk $j$. We calculate $|\tilde{a}(\omega)|^2$, simplify using $\omega_0 T \gg 1$, and assume a small range of trap frequencies to average the oscillatory terms, resulting in

$$\Delta N = \frac{1}{2}\left(\frac{6D/x_{\text{zpf}}}{\omega_0^2 T^2}\right)^2. \tag{2}$$

Several relevant insights can be gleaned from this formula. First, this expression indicates our ability to move large distances $D$ with comparably small increases in time $T$. Furthermore, to maintain a constant $\Delta N$, the movement time $T \propto \omega_0^{-3/4}$. Moreover, to perform a large number of moves $k$ for a deep circuit, we can estimate $\Delta N \propto k/T^4$, suggesting that we can increase our number of moves from, for example, 5 to 80 by slowing each move from 200 µs to 400 µs. Move speed could be further improved with different $a(t)$ profiles, but inevitably with finite resources such as trap depth, quantum speed limits will eventually prevent arbitrarily fast motion of qubits across the array[30].

We now compare equation (2) with our experimental observations. In Fig. 1d we start to observe atom loss when we move 55 µm in 200 µs under a constant negative jerk. This speed limit is consistent with our above estimates: using $\omega_0 = 2\pi \times 40$ kHz and $x_{\text{zpf}} = 38$ nm, we predict $\Delta N \approx 6$ for this move, corresponding to the onset of tangible heating at this move speed. More quantitatively, we assume a Poisson distribution with mean $N$ and variance $N$ and integrate the population above some critical $N_{\text{max}}$ upon which the atom will leave the trap. From this analysis we find atom retention is given by $\frac{1}{2}(1 + \text{erf}[(N_{\text{max}} - N)/\sqrt{2N}])$.

Extended Data Fig. 2a, b measures the atom retention as a function of move time $T$ and trap frequency $\omega_0/2\pi$. Using the functional form above, for both sets of measurements, we extract an $N_{\text{max}}$ of about 30, corresponding to adding about 30 excitations before exciting the atom out of the trap. Such a limit is physically reasonable as the absolute trap depth of 4 MHz implies only about 100 levels, the atom starts at finite temperature, and moreover the effective trap frequency reduces once the anharmonicity of the trap starts to play a role. We note that these estimates are only approximate (we roughly estimate $\omega_0$ for the trap depths used during the motion), but nonetheless suggests our motion limit is consistent with physical limits for our chosen $a(t)$. Our analysis here also neglects the acoustic lensing effects associated with ramping the AOD frequency, which causes astigmatism by focusing one axis to

a different plane and thus deforms the trap and reduces the peak trap intensity (and $\omega_0$) as given by the Strehl ratio.

Additional heating and loss during the circuit can also be caused by repeated short drops for performing two-qubit gates, where the tweezers are briefly turned off to avoid anti-trapping of the Rydberg state and light shifts of the ground–Rydberg transition. However, drop–recapture measurements in Extended Data Fig. 2c suggest that the 500-ns drops we use experimentally have a negligible effect until hundreds of drops per atom (corresponding to hundreds of CZ gates). We find that atom loss and heating as a function of number of drops are well described by a diffusion model, which would then predict that reducing atom temperature by a factor of 2× (reducing thermal velocity by $\sqrt{2}$×) and reducing the drop time $t_{\text{drop}}$ by 2×, together would increase the number of possible CZ gates per atom to thousands.

## Two-qubit CZ gates implementation

We implement our two-qubit gates and calibrations following ref. [5]. Specifically, the two-qubit CZ gate is implemented by two global Rydberg pulses, with each pulse at detuning $\Delta$ and length $\tau$, and with a phase jump $\xi$ between the two pulses. The pulse parameters are chosen such that qubit pairs, adjacent and under the Rydberg blockade constraint, will return from the Rydberg state back to the hyperfine qubit manifold with a phase depending on the state of the other qubit. The numerical values for these pulse parameters are:

$$\Delta = -0.377371\Omega$$

$$\xi = -0.621089 \times (2\pi)$$

$$\tau = 0.683201/[\Omega/(2\pi)]$$

For our experiments in Figs. 1–3, we operate with a two-photon Rydberg Rabi frequency of $\Omega/2\pi = 3.6$ MHz, giving a theoretical $\tau = 190$ ns and a theoretical $\Delta/(2\pi) = -1.36$ MHz. We choose the negative detuning sign to help minimize excitation into the $m_j = +1/2$ Rydberg state ($m_j$ denotes magnetic sublevel of the $70S_{1/2}$ Rydberg state), which is detuned by about 24 MHz under the field of 8.5 G (and experiences a three-times lower coupling to the Rydberg laser than the desired $m_j = -1/2$ state owing to reduced Clebsch–Gordan coefficients). In this work, we operate with strong blockade between adjacent qubits, with Rydberg–Rydberg interactions $V_0/2\pi$ ranging from 200 MHz to 1 GHz. In Fig. 4, we operate with $\Omega/2\pi = 4.45$ MHz for the two-qubit gates.

## Managing spurious phases during CZ gates

The two-qubit gate from ref. [5] induces both an intrinsic single-qubit phase, as well as spurious phases that are primarily induced by the differential light shift from the 420-nm laser. Under certain configurations, the 420-nm-induced differential light shift on the hyperfine qubit can be exceedingly large (>8 MHz), yielding phase accumulations on the hyperfine qubit of about 6π. Small, percent-level variations of the 420-nm intensity can thus lead to significant qubit dephasing.

Reference [5] addresses this 420-induced-phase issue by performing an echo sequence: after the CZ gate, the 1,013-nm Rydberg laser is turned off, a Raman π pulse is applied and then the 420-nm laser is pulsed again to cancel the phase induced by the 420 light during the CZ gate. This method echoes out the 420-induced phase, but comes at a cost of a factor of two increase in the 420-induced scattering error, which is the dominant source of error in our two-qubit CZ gates.

**Echo between CZ gates.** To address these various issues, here we perform a Raman π pulse between each CZ gate to echo out spurious gate-induced phases on the hyperfine qubit (Extended Data Fig. 1). This approach has several advantages. The 420-induced phase is now cancelled by pairs of CZ gates, without explicitly applying additional 420-nm pulses to echo each individual CZ gate, thereby reducing the

scattering error of the CZ gate in this work by a factor of approximately two. We estimate that this echo technique, having reduced the scattering error incurred during each gate, roughly compensates the increased scattering rate incurred by spreading our optical power over more space in 2D, thereby giving us comparable gate fidelites to the two-qubit CZ gate fidelities of ≥97.4(2)% reported in ref. [5]. Furthermore, the echo between CZ gates also cancels the intrinsic single-qubit phase of the CZ gate, removing errors in the calibration of this parameter, as well as cancelling any other gate-induced spurious single-qubit phases such as a roughly 0.01-rad phase induced by pulsing the traps off for 500 ns for the two-qubit gate (Extended Data Fig. 1). In instances where the number of CZ gates we apply is odd, we perform the echo for the final CZ gate.

**Sign of intermediate-state detuning.** To further suppress the effect of the spurious, 420-induced phase, we operate our 420-nm laser to be red-detuned (by 2 GHz) from the $6P_{3/2}$ transition. For red detunings, the light shift on the $|0\rangle$ state and the $|1\rangle$ state are of the same sign, minimizing the differential light shift, whereas for blue detunings <6.8 GHz, the light shift on the $|0\rangle$ state and the $|1\rangle$ state have opposite signs and amplify the differential light shift.

### Sensitivity to axial trap oscillations

In typical Rydberg excitation timescales with optical tweezers, the axial trap oscillation frequencies of several kilohertz are inconsequential. Here with our circuits running as long as 1.2 ms, with Rydberg pulses throughout, we find that the axial trap oscillations can have important effects. In particular, the axial oscillations cause the atoms to make oscillations in and out of the Rydberg beams: at our estimated axial temperature of about 25 μK and axial oscillation frequency of 6 kHz, we estimate an axial spread $\sqrt{\langle z^2 \rangle} \approx 1.3$ μm. For our 20-μm-waist beams, the effect of this positional spread is relatively small on the pulse parameters of the CZ gate, but can be significant on the sensitive 420-induced phase we seek to cancel by echoing out the phase induced by CZ gates separated by about 200 μs (see previous section). When using 20-μm-waist beams, and a 2.5-GHz blue detuning of our 420-nm laser, we find that the dephasing due to the axial trap oscillations is significant (Extended Data Fig. 4). To remedy this deleterious effect, we increase the beam waist of our 420-nm laser to 35 μm (while maintaining constant intensity) and change the laser frequency to be 2-GHz red-detuned, together resulting in a significant reduction in the dephasing associated with improper echoing of the 420-nm pulse.

### Bell-state preparation and fidelity

In Fig. 1, we prepare the $|\Phi^+\rangle$ Bell state in the same way that is done in ref. [5]. After initializing a pair of qubits in $|00\rangle$, we apply $X(\pi/2)$ pulse−CZ gate−$X(\pi/4)$ pulse. We calculate and plot the raw resulting fidelity of this $|\Phi^+\rangle$ Bell state as the sum of populations in $|00\rangle$ and $|11\rangle$, averaged with the fitted amplitude of parity oscillations (example in Fig. 1c), which measures the off-diagonal coherences. In Fig. 1d, upon significant loss from movement, this fidelity estimate becomes skewed because we begin measuring an artificially large population in $|11\rangle$ (as state $|1\rangle$ is detected as loss); accordingly, we estimate the $|\Phi^+\rangle$ population as two times the population of $|00\rangle$ once the population difference between $|11\rangle$ and $|00\rangle$ becomes greater than 0.1 (an arbitrary cut-off where the effects of loss start to become significant). In Fig. 1d, for moves slower than 300 μs, we achieve an average raw Bell-state fidelity after the moving of 94.8(2)%. If we do not move or attempt to preserve coherence for 500 μs (that is, if we measure immediately after preparing the Bell state), then we measure a raw Bell-state fidelity of 95.2(1)% (not plotted here), consistent with the results in ref. [5].

### Analysis of error sources

We detail here some of our measured and estimated sources of error for an entire sequence (toric code preparation in particular, our deepest circuit). We find the total single-qubit fidelity after performing the entire

sequence is roughly 96.5% for the toric code circuit, which we measure by embedding the entire experiment in a Ramsey sequence: that is, we perform a Raman π/2 pulse, do all motion and decoupling, and then do a final π/2 pulse with variable phase to measure total contrast. We are able to account for our single-qubit fidelity quantitatively as being composed of our known single-qubit errors in Extended Data Fig. 6c.

Estimated contributions to two-qubit gate error are summarized in Extended Data Fig. 6c. These estimates come from numerical simulations in QuTiP (version 4.5.0) with experimental parameters. The effects of intermediate state scattering and Rydberg decay are included via collapse operators in the Lindblad master equation solver. Other error contributions include finite-temperature random Doppler shifts and position fluctuations, as well as laser pulse-to-pulse fluctuations, all of which are simulated using classical Monte Carlo sampling of experiment parameters. Experimental parameters used for the simulations are as follows: blue and red Rabi frequencies ($\Omega_b$, $\Omega_r$) = $2\pi \times$ (160, 90) MHz, $6P_{3/2}$ intermediate state detuning of 2 GHz, intermediate state lifetime of 110 ns, $70S_{1/2}$ Rydberg state lifetime of 150 μs, Rydberg blockade energy of 500 MHz, splitting to second Rydberg state of 24 MHz, radial and axial trap frequencies ($\omega_r$, $\omega_z$) = $2\pi \times$ (40, 6) kHz, and temperature $T$ = 20 μK. We can also use this modelling to project for future performance; by assuming a 10 times increase in available 1,013-nm intensity and that atoms are cooled to a temperature of 2 uK, we project a possible CZ gate fidelity of 99.7%, beyond the surface code threshold[38,67]. Alkaline-earth atoms could also offer other routes to high-fidelity operations for QEC[68–70].

To understand how our various single-qubit and two-qubit errors contribute to our graph-state fidelities, we perform a stochastic simulation of the quantum circuit used for graph-state preparation (Extended Data Fig. 6a, b). We utilize the Clifford properties of our circuit, allowing for efficient numerical evaluation and random sampling of many possible error realizations. The simulation is performed under a realistic error model, where the rates of ambient depolarizing noise and atom loss are measured in the experiment (Extended Data Fig. 6c). The resulting stabilizer and logical qubit expectation values agree well with those measured experimentally.

### Rydberg beam shaping and homogeneity

We shape our Rydberg beams into tophats of variable size through wavefront control using the phase profile on an SLM[25]. This ability allows us to match the height of our beam profile to the experiment zone size of any given experiment, thereby maximizing our 1,013-nm light intensity and CZ gate fidelities. We optimize our Rydberg beam homogeneity until peak-to-peak inhomogenities are below <1%. To this end, we correct all aberrations up to the window of our vacuum chamber, as done in ref. [25], which yields an inhomogeneity on the atoms of several per cent that we attribute to imperfections of the final window. To further optimize the homogeneity, we empirically tune aberration corrections on the tophat through Zernike polynomial corrections to the phase profile in the SLM plane (Fourier plane). With this procedure, we reduce peak-to-peak inhomogeneities to <1% over a range of 40–50 μm in the atom plane.

### Creation and optimization of graph layouts

We outline here a description of how we optimize our graph layouts for the cluster state, Steane code, surface code and toric code preparation. Our optimization in this work is heuristic, and future work can develop appropriate algorithms for designing optimal circuits through atom spatial arrangement and AOD trajectories. Extended Data Fig. 5 shows all of the graphs we create and the process for creating them. There are several parameters we optimize for. (1) Minimize the number of parallel two-qubit gate layers. (2) Minimize the total move distance for the moving atoms. (3) Have all moving atoms in one sublattice (all graphs realized here are bipartite) to facilitate the final local rotation of one sublattice. (4) Minimize the vertical extent of the array and the number of distinct rows (to maximize 1,013-nm light intensity and

minimize sensitivity to beam inhomogeneity between the rows). (5) When ordering gates, apply two-qubit gates as early as possible in the circuit. If a gate layer induces a bit-flip ($X$ error) then that error can propagate during subsequent gates (becoming a $Z$ error on the other qubit), so gates should be in the earliest layer possible.

### Local (sublattice) hyperfine rotations

We perform local rotations in the hyperfine basis by use of our horizontally propagating 420-nm beam, which imposes a differential light of several megahertz on the hyperfine qubit and can thus be used for realizing a fast $Z$ rotation. To realize the local $Y(\pi/2)$ rotation used throughout this work, we move one sublattice of atoms out of the 420-nm beam, then apply [global $Y(\pi/4)$]–[local $Z(\pi)$]–[global $Y(\pi/4)$]. This realizes a $Y(\pi/2)$ rotation on one sublattice and a $Z(\pi)$ rotation on the other sublattice (which is inconsequential as it then commutes with the immediately following measurement in the $Z$ basis). To apply a $Y(\pi/2)$ on the other sublattice of atoms, we add an additional global $Z(\pi)$ (implemented by jumping the Raman laser phase) between the two $Y(\pi/4)$ pulses. Future experiments will benefit from an additional set of locally focused beams for performing local Raman control of hyperfine qubit states, but we find that moving atoms works so efficiently (even for moving >50 μm to move out of the 420-nm beam) that this approach is well suited for producing a high-fidelity, homogeneous rotation on roughly half the qubits.

### Local Rydberg initialization

We perform local Rydberg control to initialize the $|\mathbb{Z}_2\rangle = |rgrg...\rangle \equiv |r1r1...\rangle$ state for studying the dynamics of many-body scars. We achieve this local initialization by applying approximately 50-MHz light shifts between $|1\rangle$ and $|r\rangle$ using 810-nm tweezers generated by an SLM onto a desired subset of sites, and then apply a global Rydberg $\pi$ pulse, which excites the non-light-shifted atoms. We use this approach here to prepare every other atom in each chain into $|r\rangle$, but emphasize that as the locations of the SLM tweezers are fully programmable, this technique can be used to prepare any initial blockade-satisfying configuration of atoms in $|1\rangle$ and $|r\rangle$.

The 50-MHz biasing light shift is significantly larger than the Rydberg Rabi frequency $\Omega/2\pi = 4.45$ MHz, leading to a Rydberg population on undesired sites of <1%. The $t = 0$ time point of Extended Data Fig. 10b shows the high-fidelity preparation of the $|\mathbb{Z}_2\rangle$ state using this approach. We note that with 810-nm light, even though the achieved biasing light shift is significant, the Raman-scattering-induced $T_1$ (of the hyperfine qubit) is still about 50 ms and thus leads to a scattering error ≲$4 \times 10^{-6}$ during the 200-ns pulse of the light-shifting tweezers. There can also be a motional effect from the biasing tweezers, with an estimated radial trapping frequency of 150 kHz, which we also deem to be negligible during the 200-ns pulse.

### Rydberg Hamiltonian

In Fig. 4, we study dynamics under the many-body Rydberg Hamiltonian

$$\frac{H_{\mathrm{Ryd}}}{\hbar} = \frac{\Omega}{2}\sum_i \sigma_i^x - \Delta\sum_i n_i + \sum_{i<j} V_{ij} n_i n_j, \tag{3}$$

where $\hbar$ is the reduced Planck constant, $\Omega$ is the Rabi frequency, $\Delta$ is the laser frequency detuning, $n_i = |r_i\rangle\langle r_i|$ is the projector onto the Rydberg state at site $i$ and $\sigma_i^x = |1_i\rangle\langle r_i| + |r_i\rangle\langle 1_i|$ flips the atomic state. For the entanglement entropy measurements in this work, we choose lattice spacings where the nearest-neighbour interaction $V_0 > \Omega$ results in the Rydberg blockade, preventing adjacent atoms from simultaneously occupying $|r\rangle$. In particular, the many-body experiments are performed on eight-atom chains, quenching to a time-independent $H_{\mathrm{Ryd}}$ with $V_0/2\pi = 20$ MHz, $\Omega/2\pi = 3.1$ MHz and $\Delta/2\pi = 0.3$ MHz. Quenching to small, positive $\Delta = 0.0173 V_0$ partially suppresses the always-positive long-range interactions and thereby is optimal for scar lifetime, as derived and shown experimentally in ref. [71].

### Coherent mapping protocol

As described in the text, we implement a coherent mapping protocol to transfer a generic many-body state in the $\{|1\rangle, |r\rangle\}$ basis to the long-lived and non-interacting $\{|0\rangle, |1\rangle\}$ basis. To achieve this mapping, immediately following the Rydberg dynamics we apply a Raman $\pi$ pulse to map $|1\rangle \to |0\rangle$, and then a subsequent Rydberg $\pi$ pulse to map $|r\rangle \to |1\rangle$ (ref. [72]).

Even for perfect Raman and Rydberg $\pi$ pulses (on isolated atoms), there are three key sources of infidelity associated with this mapping process. (1) Any population in blockade-violating states (that is, two adjacent atoms both in $|r\rangle$) will be strongly shifted off-resonance for the final Rydberg $\pi$ pulse. As such, this atomic population will be left in the Rydberg state and lost. (2) Long-range interactions, for example, from next-nearest neighbours, will detune the final Rydberg $\pi$ pulse from resonance and thus reduce pulse fidelity. As the long-range interactions are not the same for all many-body microstates, this effect cannot be mitigated by a simple shift of the detuning. (3) Dephasing of the state occurs throughout the duration of the Raman $\pi$ pulse, predominantly from Doppler shifts between the ground states $|0\rangle$ and $|1\rangle$ and the Rydberg state $|r\rangle$. Although these random on-site detunings are also present during the many-body dynamics, turning the Rydberg drive $\Omega$ off allows the system to freely accumulate phase and makes us particularly sensitive to dephasing errors.

We now detail our mitigation of the above error mechanisms. To minimize errors from (1), we perform our many-body dynamics with $\Omega^2/(2V_0^2) \approx 0.01$. This minimizes the probability of an atom to violate blockade to be of order 1%. To help minimize errors from (2), we increase the amplitude of the 420-nm laser for the final $\pi$ pulse by a factor of 2×, such that $(V_{\mathrm{NNN}}/\Omega)^2 = 0.005$ (where $V_{\mathrm{NNN}}$ is the interactions with next-nearest neighbours), reducing pulse errors from long-range interactions to order 1%. Finally, to reduce errors from (3), we perform a fast Raman $\pi$ pulse and leave only 150 ns between ending the many-body Rydberg dynamics and beginning the Rydberg $\pi$ pulse. The 150-ns gap is comparably short relative to the $T_2^* \approx 3$–4 μs of the $\{|g\rangle, |r\rangle\}$ basis, leading to a random phase accumulation of the order of about $0.02 \times 2\pi$ rad per particle, but is further compounded by having entangled states of $N$ particles in one copy accumulating a random phase relative to entangled states of $N$ particles in the second copy. We study these various effects numerically in Extended Data Fig. 9c.

Finally, we note that the global Raman beam induces a light-shift-induced phase shift of about $\pi$ on $|0\rangle$ and $|1\rangle$ relative to $|r\rangle$ during the Raman $\pi$ pulse. Similarly, the global 420-nm laser also induces a light-shift-induced phase shift of about $\pi$ between $|0\rangle$ and $|1\rangle$ during the Rydberg $\pi$ pulse. Although the measurements we perform here are interferometric (in other words, the singlet state we measure is invariant under global rotations) and thus not affected by these global phase shifts, in future work these phase shifts can be measured and accounted for where relevant.

### Measuring entanglement entropy

The second-order Renyi entanglement entropy is given by $S_2(\mathrm{A}) = -\log_2 \mathrm{Tr}\left[\rho_\mathrm{A}^2\right]$, where $\mathrm{Tr}\left[\rho_\mathrm{A}^2\right]$ is the state purity of reduced density matrix $\rho_\mathrm{A}$ on subsystem A. The purity can be measured with two copies by noticing that $\mathrm{Tr}\left[\rho_\mathrm{A}^2\right] = \mathrm{Tr}\left[\hat{\mathbf{S}}\rho_\mathrm{A} \otimes \rho_\mathrm{A}\right]$ is the expectation value of the many-body SWAP operator $\hat{\mathbf{S}}$ (ref. [10,11]). The many-body SWAP operator is composed of individual SWAP operators $\hat{s}_i$ on each twin pair, that is, $\hat{\mathbf{S}} = \Pi_i \hat{s}_i$ (with $i \in \mathrm{A}$). Measuring this expectation value amounts to probing occurences of the singlet state $\frac{|01\rangle - |10\rangle}{\sqrt{2}}$ (with eigenvalue −1 under $\hat{s}_i$), as all other $\hat{s}_i$ eigenstates have eigenvalue +1. Occurences of the singlet state in each twin pair, that is, the Bell state $|\Psi^-\rangle$, is extracted by a Bell measurement circuit (with an additional local $Z(\pi)$, see next paragraph), which maps $|\Psi^-\rangle \to |00\rangle$ and can thereafter be measured in the computational basis. As such, after performing the Bell measurement circuit, we analyse the resulting bit-string outputs and calculate the purity of any subsystem A by calculating $\langle \Pi_{i\in\mathrm{A}} \hat{s}_i \rangle$: that is, we measure purity as

the average parity = $\langle(-1)^{observed\ |00\rangle\ pairs}\rangle$ within A. In the absence of experimental imperfections, the purity will equal 1 for the whole system, and be less than 1 for subsystems that are entangled with the rest of the system.

A Bell measurement circuit can be decomposed into applying an $X(\pi/2)$ rotation on one atom of the twin pair, then applying a CZ gate and then a global $X(\pi/2)$ rotation. In other measurements, we realize a local $X(\pi/2)$ by doing a global $X(\pi/4)$ rotation, then local $Z(\pi)$ rotation and then global $X(\pi/4)$. However, we note that for this singlet measurement circuit, the first $X(\pi/4)$ is redundant as the singlet state is invariant under global rotations, and so for the local $X(\pi/2)$ we only apply the local $Z(\pi)$ and then the second global $X(\pi/4)$. This effectively realizes the $X(\pi/2)$ on one qubit, up to a $Z(\pi)$ on the other qubit (not shown in circuit diagram in Fig. 4). Under this simplification, the Bell measurement circuit to map $|\Psi^-\rangle \to |00\rangle$ can be roughly understood as the reverse of the Bell-state preparation circuit in ref. [5], which is precisely how we calibrate the parameters of the Bell measurement.

**Calibrating and benchmarking the interferometry.** To validate the interferometry measurement (and check for proper calibration), we benchmark it separately from the many-body dynamics and coherent mapping protocol. We perform this benchmarking by preparing independent qubits in identical, variable single-qubit superpositions (through a global Raman pulse of variable time) and ensuring that the interferometry rarely results in $|00\rangle$ for all the variable initial product states (Extended Data Fig. 8a). We find this is an important benchmarking step, because we find that small miscalibrations of the Bell measurement can lead to lower fidelity (that is, higher entropy) for different initial product states and thereby result in additional spurious signals in an entanglement entropy measurement. We note that this measurement is particularly sensitive to the single-qubit phase immediately before the final $X(\pi/2)$ pulse (induced by the CZ gate and cancelled by a global $Z(\theta)$ pulse).

**Additional many-body data and details**
To benchmark our method of measuring entanglement entropy in a many-body system, in Extended Data Fig. 8b we study the entanglement dynamics after initializing two proximal atoms in $|1\rangle$ and resonantly exciting to the Rydberg state for a variable time $t$. Under conditions of Rydberg blockade, this excitation results in two-particle Rabi oscillations between $|11\rangle$ and the entangled state $|W\rangle = \frac{1}{\sqrt{2}}(|1r\rangle + |r1\rangle)$ (top panel of Extended Data Fig. 8b)[3,13,72]. The state purity of this two-particle system is measured by performing Bell measurements on atom pairs from two identical copies. Locally, the measured purity of the one-particle subsystem reduces to a value of about 0.5 when the system enters the maximally entangled $|W\rangle$ state, at which point the reduced density matrix of each individual atom is maximally mixed. In contrast, the purity of the global, two-particle state remains high. The observation that the global-state purity is higher than the local-subsystem purity is a distinct signature of quantum entanglement[11,12].

For the data shown in Fig. 4c, e, we subtract the data by an extensive classical entropy as is done in ref. [12]. This fixed, time-independent offset is given by the entropy per particle, that is, (global entropy at quench time $t = 0$) × (subsystem size)/(global system size). In Extended Data Fig. 9a, we show the raw entanglement entropy measurements alongside numerics, to indicate the size of the extensive classical entropy contribution. In plotting, we also delay the theory curves by 10 ns to account for the fact that the Raman $\pi$ pulse cuts off the final 10 ns of the Rydberg evolution, which is done to keep the coherent mapping gap as short as possible and minimize Doppler dephasing. Furthermore, in Extended Data Fig. 9b we plot the measured global purity and compare it with numerical simulations incorporating experimental errors (Extended Data Fig. 9c).

In Extended Data Fig. 10, we show additional many-body data on the eight-atom chain system, with the same parameters as those used in the main text. We show the measured single-site entropy of each site in the eight-atom chain for the $|\mathbb{Z}_2\rangle$ quench in Extended Data Fig. 10a. Furthermore, in Extended Data Fig. 10b, we plot the global Rydberg population, measured in both the $\{|1\rangle, |r\rangle\}$ basis and the $\{|0\rangle, |1\rangle\}$ basis.

## Data availability
The data that supports the findings of this study are available from the corresponding author on reasonable request.

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

**Acknowledgements** We thank I. Cong and X. Gao for their insight into QEC and for pointing us to the connections between graph states and QEC codes; M. Abobeih, H. Bernien, M. Cain, S. Choi, M. Endres, S. Evered, W. W. Ho, T. Manovitz, A. Omran, M. Serbyn and H. Zhou for reading of the manuscript and helpful discussions. We acknowledge financial support from the Center for Ultracold Atoms, the National Science Foundation, the Vannevar Bush Faculty Fellowship, the US Department of Energy (DE-SC0021013 and DOE Quantum Systems Accelerator Center, contract number 7568717), the Office of Naval Research, the Army Research Office MURI (grant number W911NF-20-1-0082) and the DARPA ONISQ programme (grant number W911NF2010021). D.B. acknowledges support from the NSF Graduate Research Fellowship Program (grant DGE1745303) and The Fannie and John Hertz Foundation. H.L. acknowledges support from the National Defense Science and Engineering Graduate (NDSEG) fellowship. G.S. acknowledges support from a fellowship from the Max Planck/Harvard Research Center for Quantum Optics. N.M. acknowledges support by the Department of Energy Computational Science Graduate Fellowship under award number DE-SC0021110. H.P. acknowledges support by the Army Research Office (grant number W911NF-21-1-0367)

**Author contributions** D.B., H.L., G.S., T.T.W., S.E., M.K. and A.K. contributed to building the experimental setup, performed the measurements and analysed the data. M.K., N.M. and H.P. performed theoretical analysis. All work was supervised by M.G., V.V. and M.D.L. All authors discussed the results and contributed to the manuscript.

**Competing interests** M.G., V.V. and M.D.L. are co-founders and shareholders of QuEra Computing. A.K. is an executive at and shareholder of QuEra Computing. All other authors declare no competing interests.

**Additional information**
**Correspondence and requests for materials** should be addressed to Mikhail D. Lukin.

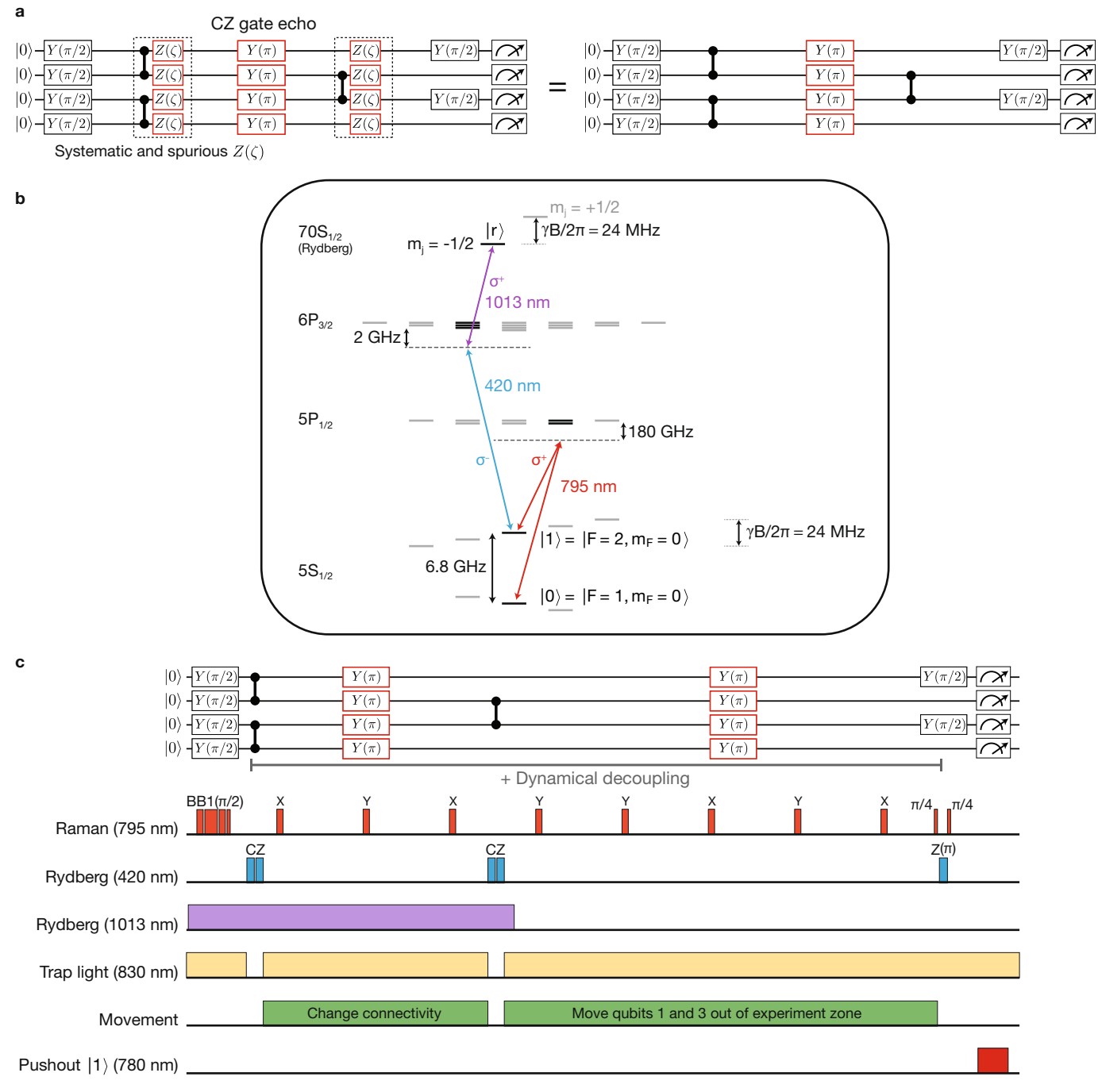

**Extended Data Fig. 1 | CZ gate echo, atomic level structure, and typical pulse sequence. a**, The two-qubit gates we apply, in addition to applying a controlled-Z operation between the two qubits, also induce a single-qubit phase $Z(\zeta)$ to both qubits, composed of the intrinsic phase of the CZ gate[5] and additional spurious phases from the 420-nm Rydberg laser and pulsing the traps off. Since we apply all gates in parallel by global pulses of the Rydberg laser, if a qubit is not adjacent to another qubit, it does not perform a CZ gate but still acquires the same $Z(\zeta)$ (identical to being adjacent to another qubit in state $|0\rangle$, which is dark to the Rydberg laser). As diagrammed, we cancel the additional, undesired $Z(\zeta)$ by applying a $\pi$ pulse between pairs of CZ gates.

This echo procedure removes any need to calibrate the intrinsic phase from the CZ gate, and renders us insensitive to any spurious changes in $Z(\zeta)$ slower than ~200 μs. The additional $Y(\pi)$ propagates in a known way through the CZ gates and multiplies certain stabilizers by a −1 sign, which simply redefines the sign of stabilizers and logical qubits. **b**, Level diagram showing key [87]Rb atomic levels used. Our Rydberg excitation scheme from $|1\rangle$ to $|r\rangle$ is composed of a two-photon transition driven by a 420-nm laser and a 1013-nm laser (see ref. [25] for description of laser system). A DC magnetic field of $B = 8.5$ G is applied throughout this work. **c**, A typical pulse sequence for running a quantum circuit.

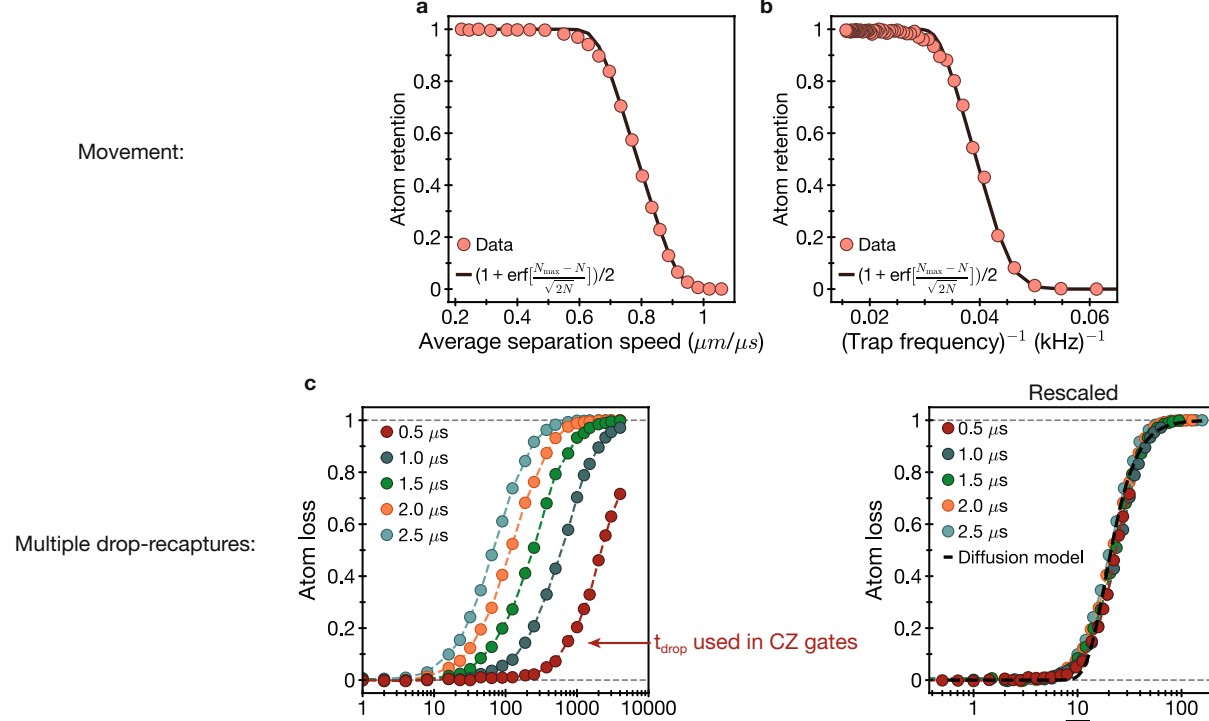

Movement:

Multiple drop-recaptures:

**Extended Data Fig. 2 | Movement characterization and multiple drop-recaptures. a**, Atom retention as a function of average separation speed $2D/T$ (as is plotted in Fig. 1d of the main text for separating Bell pairs), with subtracted background loss of 0.7%. The inset in Fig. 1d of the main text is normalized by (Atom retention)$^2$ (without subtracting background loss). Dark curve is calculated using experimental parameters and Eq. 2, matched to the experimental data by setting $N_{max} = 26$ and averaging over a range of $\omega_0/2\pi$ of ±15% around an average $\omega_0/2\pi = 40$ kHz. **b**, Atom retention as a function of inverse trap frequency $(2\pi/\omega_0)$ after the four moves of the surface code circuit. For calculating the atom loss here we set $N_{max} = 33$ and average the trap frequencies over a range of ±15%. We note that these quantitative estimates are sensitive to $\omega_0$ which we roughly estimate. **c**, Atom loss as a function of drop time and number of drop loops, with 100 $\mu$s wait between each drop. When running quantum circuits we use 500-ns drops for each CZ gate (to avoid anti-trapping of the Rydberg state and light shifts of the transition), for which we observe here that hundreds of drops can be made (corresponding to hundreds of possible CZ gates per atom) before atom loss becomes significant. **d**, By rescaling the x-axis of the data to $t_{drop}\sqrt{N}$, we find the data of the various $t_{drop}$ collapse onto a universal curve, suggesting a diffusion model for explaining the atom loss after a certain number of drops. By modeling such a diffusion process analytically we obtain the black curve with a temperature of 10 $\mu$K and a trapping radius of 1 $\mu$m.

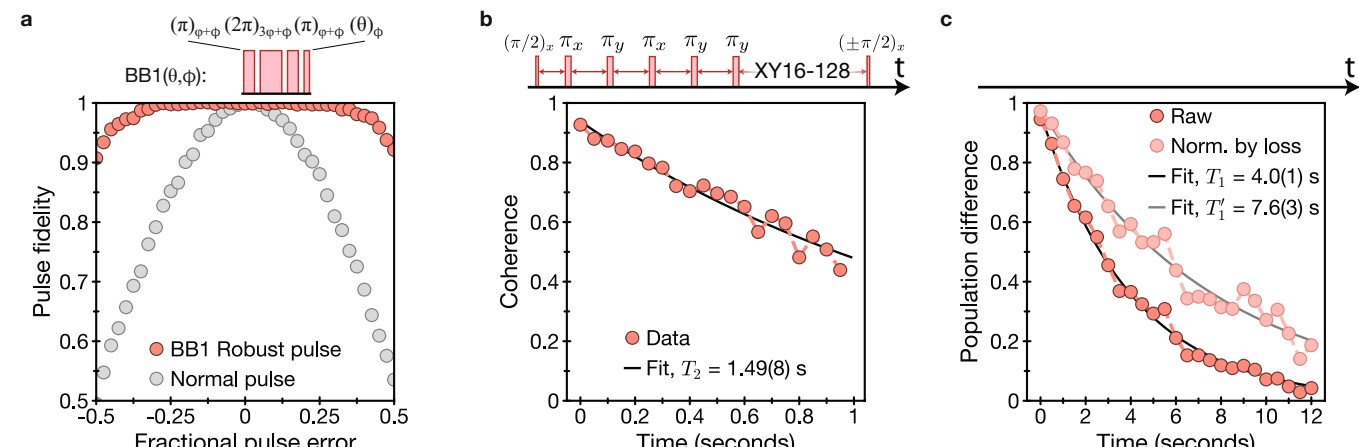

**Extended Data Fig. 3 | Robust single-qubit control and qubit coherence.**
**a**, Robust BB1 single-qubit rotation in comparison to a normal single-qubit rotation, as a function of pulse area error. An arbitrary BB1($\theta, \phi$) rotation on the Bloch sphere of angle $\theta$ about axis $\phi$ is realized with a sequence of four pulses: $(\pi)_{\varphi+\phi}(2\pi)_{3\varphi+\phi}(\pi)_{\varphi+\phi}(\theta)_\phi$, where $\varphi = \cos^{-1}(-\theta/4\pi)$[60]. Pulse fidelity is measured here for a $\pi$ pulse, defined such that the fidelity is the probability of successful transfer from $|0\rangle \rightarrow |1\rangle$, including SPAM correction. **b**, Preserving hyperfine qubit coherence using dynamical decoupling (XY16 with 128 total $\pi$ pulses).

Qubit coherence is observed on a timescale of seconds, with a fitted coherence time $T_2 = 1.49(8)$s. Data is measured with either a $+\pi/2$ or $-\pi/2$ pulse at the end of the sequence, and these curves are then subtracted to yield the coherence y-axis. **c**, Hyperfine qubit $T_1$, measured by the difference of final $F = 2$ populations between measurements starting in $|F = 2, m_F = 0\rangle$ and $|F = 1, m_F = 0\rangle$. Atom loss without cooling is separately measured (predominantly arising from vacuum loss) and normalized to also measure the intrinsic spin relaxation time $T_1'$ in the absence of atom loss. All data here is measured in 830-nm traps.

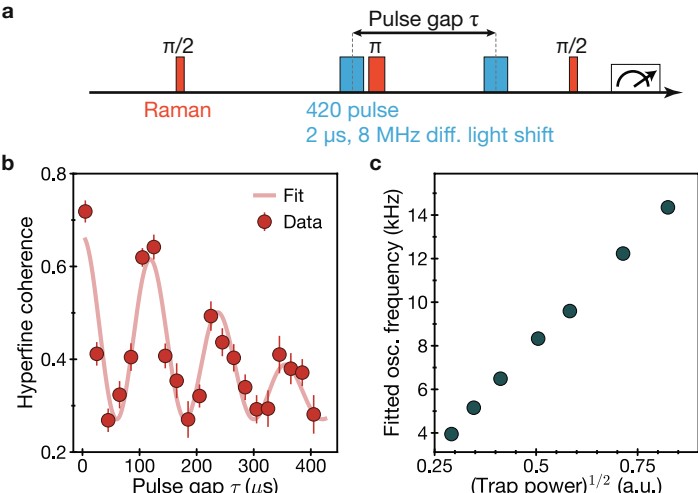

**Extended Data Fig. 4 | Effect of axial trap oscillations on echo fidelity of 420-nm Rydberg pulse. a**, Noise correlation measurement of the 420-nm Rydberg laser pulse intensity. In the blue-detuned configuration used in this figure only, the 420-nm laser induces an 8 MHz differential light shift on the hyperfine qubit, and consequently a phase accumulation of $32\pi$ during a 2-$\mu$s pulse (our CZ gates are 400-ns total). Small fluctuations of the 420-nm laser intensity lead to large fluctuations in phase accumulation of the hyperfine qubit, and thus cause significant dephasing. The echo sequence diagrammed here probes the correlation of the accumulated phase between two 420-nm pulses separated by a variable time $\tau$, and thus informs how far-separated in time the 420-nm pulses can be while still properly echoing out fluctuations in the 420-nm intensity. **b**, Hyperfine coherence (a proxy for echo fidelity) versus gap time $\tau$ between the two 420-nm pulses. The echo fidelity decreases initially due

to a decorrelation of the 420-nm intensity, but then increases again, showing that the correlation of the 420-nm intensity is non-monotonic. The decaying oscillations are fit to a functional form of $y = y_0 + A\cos^2(\pi f \tau)\exp[-(\tau/T)^2]$. **c**, The fitted oscillation frequency $f$ of the correlation/decorrelation of the noise follows a square-root relationship with the trap power, and is consistent with the expected axial trap oscillation frequency. These observations indicate that a significant portion of the correlation/decorrelation of the 420-nm noise arises from the several-$\mu$m axial oscillations of the atom in the trap. For this measurement, we intentionally displace the 420-nm beam by several $\mu$m in order to place the atom on a slope of the beam, increasing our sensitivity to this phenomenon. For the other experiments in our work, we minimize sensitivity to these effects by operating in the center of a larger (35-micron-waist) 420-nm beam and operating red-detuned of the intermediate-state transition.

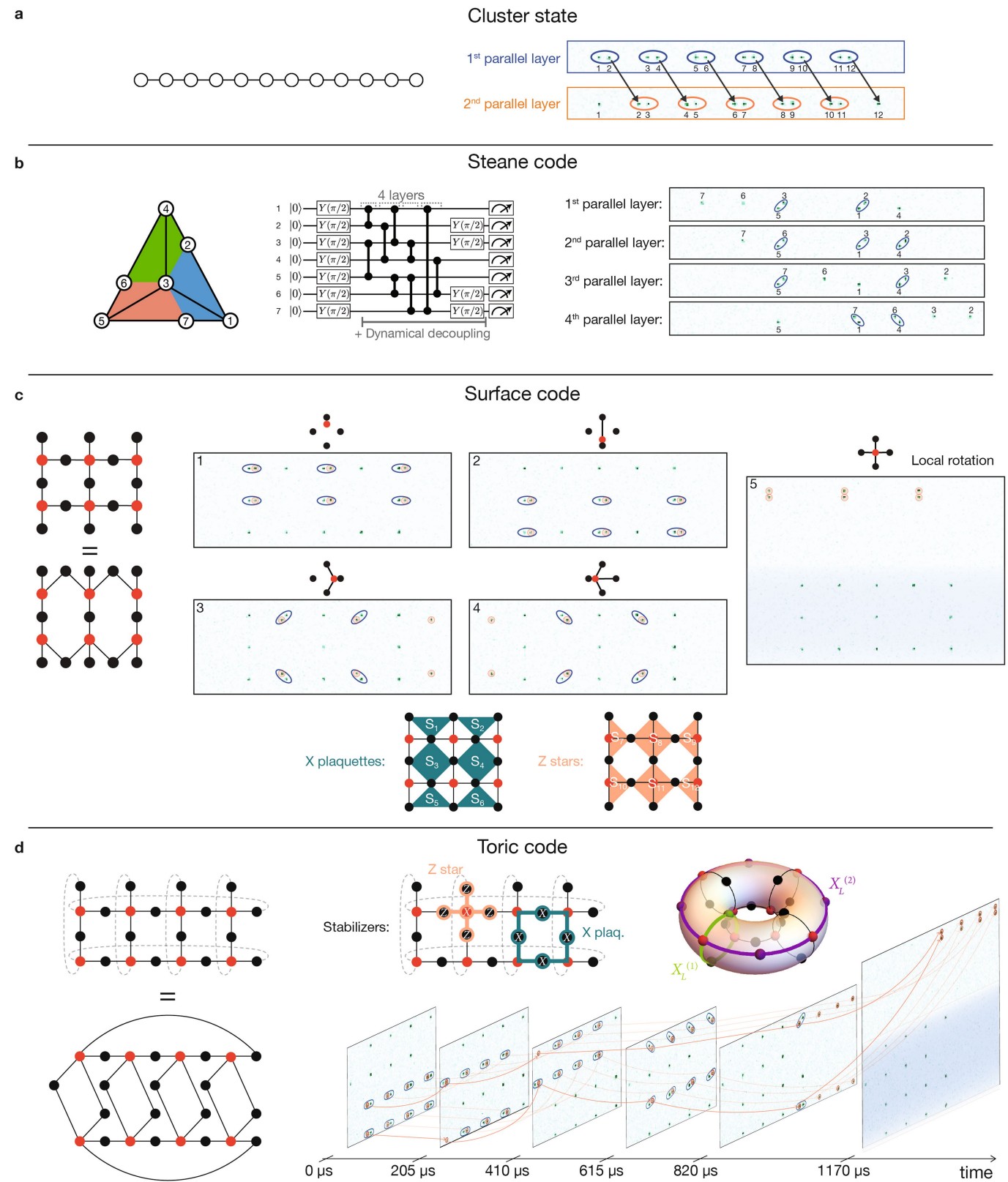

**Extended Data Fig. 5 | Movement schematics.** Schematics showing the gate-by-gate creation of **a** the 1D cluster state, **b** the Steane code, **c** the surface code, and **d** the toric code (see also Supplementary Video 1), in a side-by-side comparison. These various graph states are all generated in the same way, and encoding a desired circuit is a matter of positioning the atoms in different initial positions and applying an appropriate AOD waveform. To realize a desired circuit, atom layouts and trajectories are optimized heuristically in the way described in the Methods text. Panel c also shows the definition of surface code stabilizers as ordered in the main text.

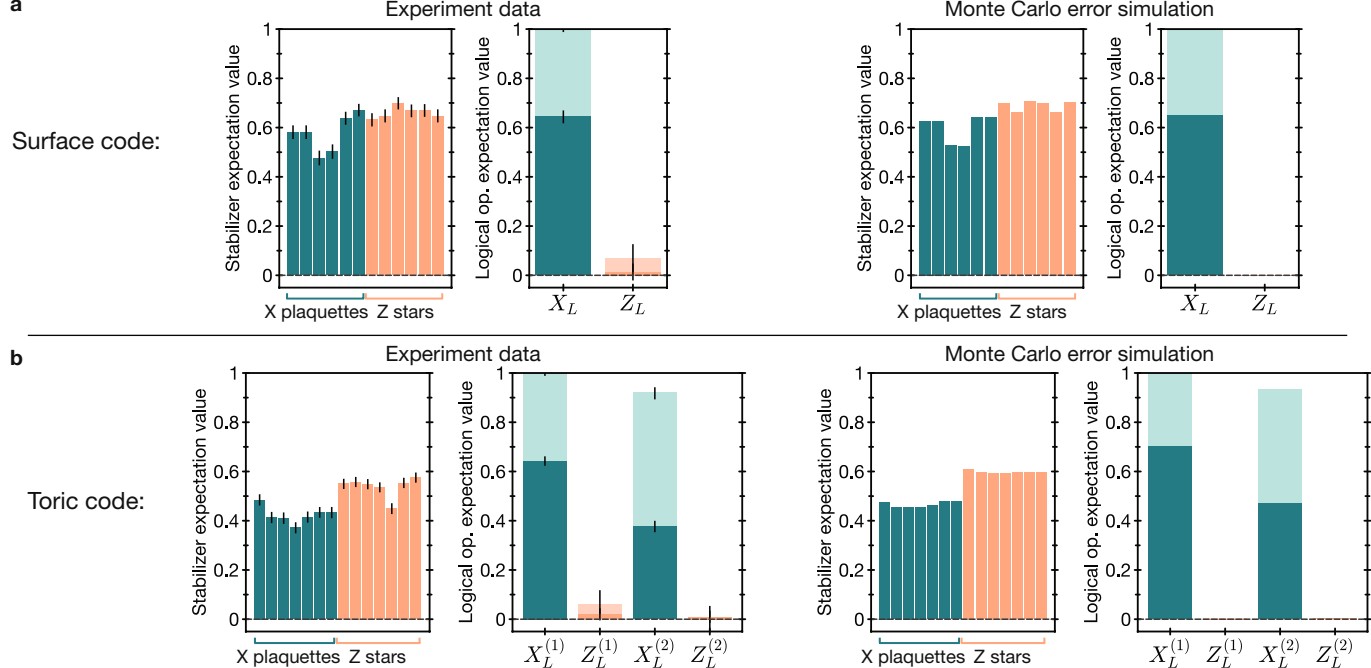

**a** Surface code:

Experiment data — Monte Carlo error simulation

**b** Toric code:

Experiment data — Monte Carlo error simulation

**c**

*Single-qubit errors (quoted numbers are for 1.2-ms toric code circuit with 5 layers).*

| Error source | Estimated error | Physical origin |
|---|---|---|
| $|1\rangle$ misidentified as $|0\rangle$ | 0.2-0.3% | Scattering from $F' = 2$ during pushout on $F' = 3$ |
| $|0\rangle$ misidentified as $|1\rangle$ | 0.1-0.2% | 10-s vacuum lifetime |
| Atom lost before circuit begins | 0.4-0.5% | 10-s vacuum lifetime |
| Pumping into $|0\rangle$ fails | 0.1-0.2% | Imperfect $m_F = \pm 1$ pulses for opt. pump. |
| Atom loss from movement | <0.1% | See ED Fig. 2 |
| Total pulse errors (including D.D.) | 0.5-1.0% | Pulse errors and Raman scattering |
| Ambient dephasing (Z) errors | 1-2% | Hyperfine $T_2$ |
| Ambient bit-flip (X) errors | <0.1% | Hyperfine $T_1$ |
| Measured total single-qubit fidelity | 96.5% | |

*Two-qubit errors.*

| Error source & physical origin | Estimated error |
|---|---|
| Intermediate state scattering | 1.1% (simulated) |
| Finite temperature (Doppler) | 0.3% (simulated) |
| Finite Rydberg lifetime | 0.2% (simulated) |
| 1% shot-to-shot power fluctuations | < 0.1% (simulated) |
| Atom loss (0.2-0.3% sim., included already in other error sources) | 1% (measured) |
| Pulse rise-time, shape imperfections, and miscalibrations | Unaccounted for |
| Estimated fidelity from accounted-for, simulated errors | ≈ 98.3% |
| Estimated fidelity by preparing Bell states (SPAM-corrected) | ≈ 97.4% |
| Estimated fidelity from MC simulations of graph-state fidelity | ≈ 97.2% |

**Extended Data Fig. 6 | Error simulations and tabulated single-qubit and two-qubit error estimates.** We compare our measured graph state fidelities to those from a stochastic Monte Carlo simulation of our system for **a**, the surface code and **b**, the toric code. We find that the simulated stabilizers agree well with the experimental data for this empirical depolarizing noise model. In addition, for the surface code (toric code) in the experiment we find 35% (20%) of measurements detect no stabilizer errors, compared to 40% (26%) in the simulation. We assume two-qubit errors are described by rates of 0.2% Y error, 0.2% X error, 0.5% Z error, and 0.5% loss per qubit per parallel layer (4 layers for surface code, 5 layers for toric code), corresponding to a 97.2% CZ-gate fidelity. We also add ambient, single-qubit errors at a rate of 0.1% Y error, 0.1% X error, 0.4% Z error, and 0.2% loss per qubit per parallel layer, as well as an initial 1% loss before the circuit begins (empirically factoring in SPAM errors). **c**, Tabulation of single-qubit (SQ) and two-qubit (TQ) gate errors that are measured, estimated, and extrapolated. Simulated TQ fidelities include the 0.6% scattering error from the 420-nm echo pulse. The estimated TQ fidelities are given for the experiments of the surface code and toric code, but is an underestimate of the TQ fidelities for the cluster state and Steane code measurements where we increase the 1013-nm intensity by 2× and reduce the 420-nm intensity by 2×, increasing gate fidelity. The Bell state estimate of CZ gate fidelity is similarly done with 2× higher 1013 intensity, but includes the 420-nm echo pulse, and consequently yields a similar gate fidelity as the surface and toric code estimates.

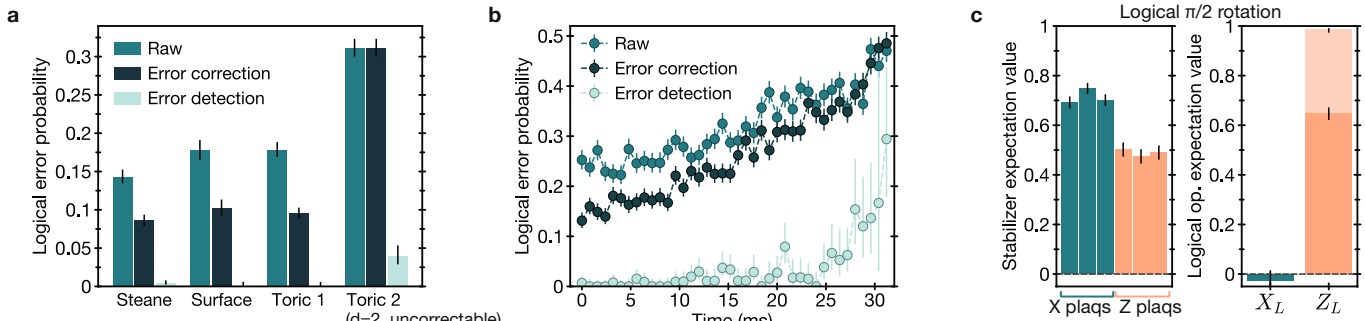

**Extended Data Fig. 7 | Properties of encoded logical states. a**, Summary of logical error probabilities for the various error correcting graphs made in this work (all in logical state $|+\rangle_L$), for raw measurements as well as implementing error correction and error detection in postprocessing. Error correction for the Steane code is implemented with the Steane code decoder[36,73] and is implemented with the minimum-weight-perfect-matching algorithm for the surface and toric codes[38]. For the even-distance toric code, when correction is ambiguous we do not flip the logical qubit, and accordingly the distance $d = 2$ logical qubit does not change under the correction procedure. We remark that the observed fidelities are comparable to similar demonstrations in state-of-the-art experiments with other platforms[8,74]. These will need to be improved to surpass the threshold for practical error correction[38] (see Methods text). **b**, Lifetime of the logical $|+\rangle_L$ state on the surface code, with correction and detection performed in postprocessing as in **a**. After state preparation, the $|+\rangle_L$ state is held for a variable time before projective measurement, with two $\pi$ pulses applied for dynamical decoupling (lifetime can be extended significantly further by applying e.g. 128 $\pi$ pulses as done in Extended Data Fig. 3b). Some experimental parameters are slightly different here compared to those in **a**, hence the higher error rates here at the time 0 point. **c**, Logical $\pi/2$ rotation on the Steane code to prepare logical qubit state $|0_L\rangle$. The Steane code, surface code, and toric code all have transversal single-qubit Clifford operations on the logical qubit[8,36] (including in-software rotations of the lattice), which is a high-fidelity operation in our system since the transversal rotations are implemented in parallel with our global Raman laser and the physical single-qubit fidelities are high. We show a logical $\pi/2$ rotation here for the Steane code as an example but emphasize that we can readily realize the various basis states for all of these codes.

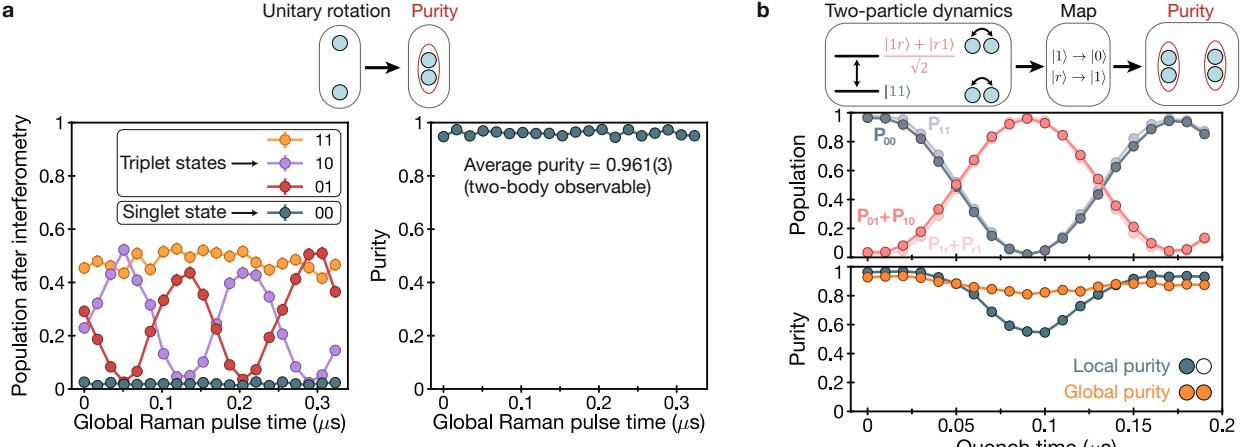

**Extended Data Fig. 8 | Benchmarking the interferometry measurement.**
**a**, To benchmark our gate-based interferometry technique, we prepare variable single-particle pure states (by applying a variable-length resonant Raman pulse) and then reconfigure the system and apply the interferometry circuit on twin pairs. The interferometry circuit converts the anti-symmetric singlet state $|\Psi^-\rangle$ to the computational basis state $|00\rangle$, while converting the symmetric triplet states to other computational states. We plot the resulting twin pair output states in the left panel. We rarely observe the $|00\rangle$ state (1.95(2)% of measurements), with a measurement fidelity independent of the initial state. This low probability $P_{00}$ of observing $|00\rangle$ corresponds to a high extracted single-particle purity of $2P_{00} - 1 = 0.961(3)$ (right panel). We find this measurement to be a useful benchmark, as interferometry miscalibrations can result in significant state-dependence of the observed purity that would then compromise the validity of the many-body entanglement entropy measurement. **b**, Benchmarking the entanglement entropy measurement with Bell state arrays. (Top) Microstate populations during two-particle oscillations between $|11\rangle$ and $\frac{1}{\sqrt{2}}(|1r\rangle + |r1\rangle)$ under a Rydberg pulse of variable duration. Faint lines show measurement results in the $\{|1\rangle, |r\rangle\}$ basis, and dark lines show results in the $\{|0\rangle, |1\rangle\}$ basis after the coherent mapping process. (Bottom) Measured local and global purities by analyzing the number parity of observed $|00\rangle$ twin pairs in each measurement. For this two-particle data we use a gap of 230 ns in the coherent mapping sequence as opposed to the 150-ns gap used in the many-body data.

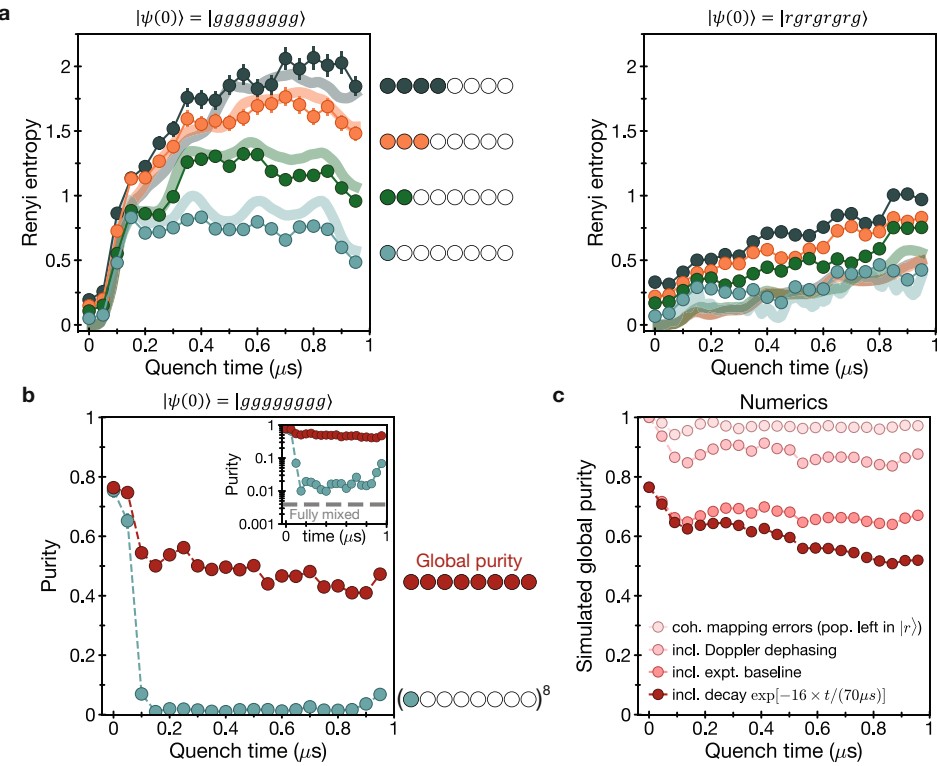

**Extended Data Fig. 9 | Raw many-body data and numerical modeling of errors. a**, Raw measured Renyi entropy without subtracting the extensive classical entropy, as a function of subsystem size for quenches from $|rgrgrgrg\rangle$ and $|gggggggg\rangle$. The Renyi entropy of the 4-atom subsystem is the same underlying data as the half-chain entanglement entropy plotted in Fig. 4d of the main text. In the main text, we subtract the data by a fixed offset given by the classical entropy-per-particle, corresponding to the time = 0 offset for each subsystem size. The extensive, classical entropy offset is slightly larger for the $|rgrgrgrg\rangle$ quench due to non-unity fidelities both of preparing $|r\rangle$ and mapping $|r\rangle \rightarrow |1\rangle$. **b**, Raw global purity after the $|gggggggg\rangle$ quench. The global purity is a sensitive proxy for the fidelity of our entire process. We find this 16-body observable, composed of three-level systems, remains > 100× the purity expected for a fully mixed state of 8 qubits $(1/2^8)$ (see inset). For comparison of scale we also plot single-particle purity to the 8th power, to indicate what the global purity would be if the measurement results on each twin were uncorrelated. **c**, Global purity for the 8-atom quench calculated through

numerical modeling of the three-level system $\{|0\rangle, |1\rangle \equiv |g\rangle, |r\rangle\}$ with a variety of simulated error sources. We model the experimentally measured purity by calculating the expectation value of the SWAP operator in the $\{|0\rangle, |1\rangle\}$ basis between two independent chains, taking into account that residual population in $|r\rangle$ results in atom loss and measurement associated with the +1 eigenvalue of the SWAP operator (as the twin state $|00\rangle$ can no longer be detected). The top curve includes only errors from population left in $|r\rangle$ following the coherent mapping step (see methods text). The second-from-top curve includes single-site dephasing ($T_2^*$) during the Rydberg dynamics and the coherent mapping gap, modeled by a random on-site detuning which is Gaussian-distributed with zero mean and standard deviation of 100 kHz. The third and fourth curves include multiplication by the experimentally observed raw global purity at quench time $t = 0$, and then further multiplying empirically by an exponential decay $\exp[-16 \times t/(70\,\mu s)]$ as a simple model for scattering and decay errors with an experimentally estimated rate of roughly 70 $\mu$s for each of the 16 atoms between the two chains.

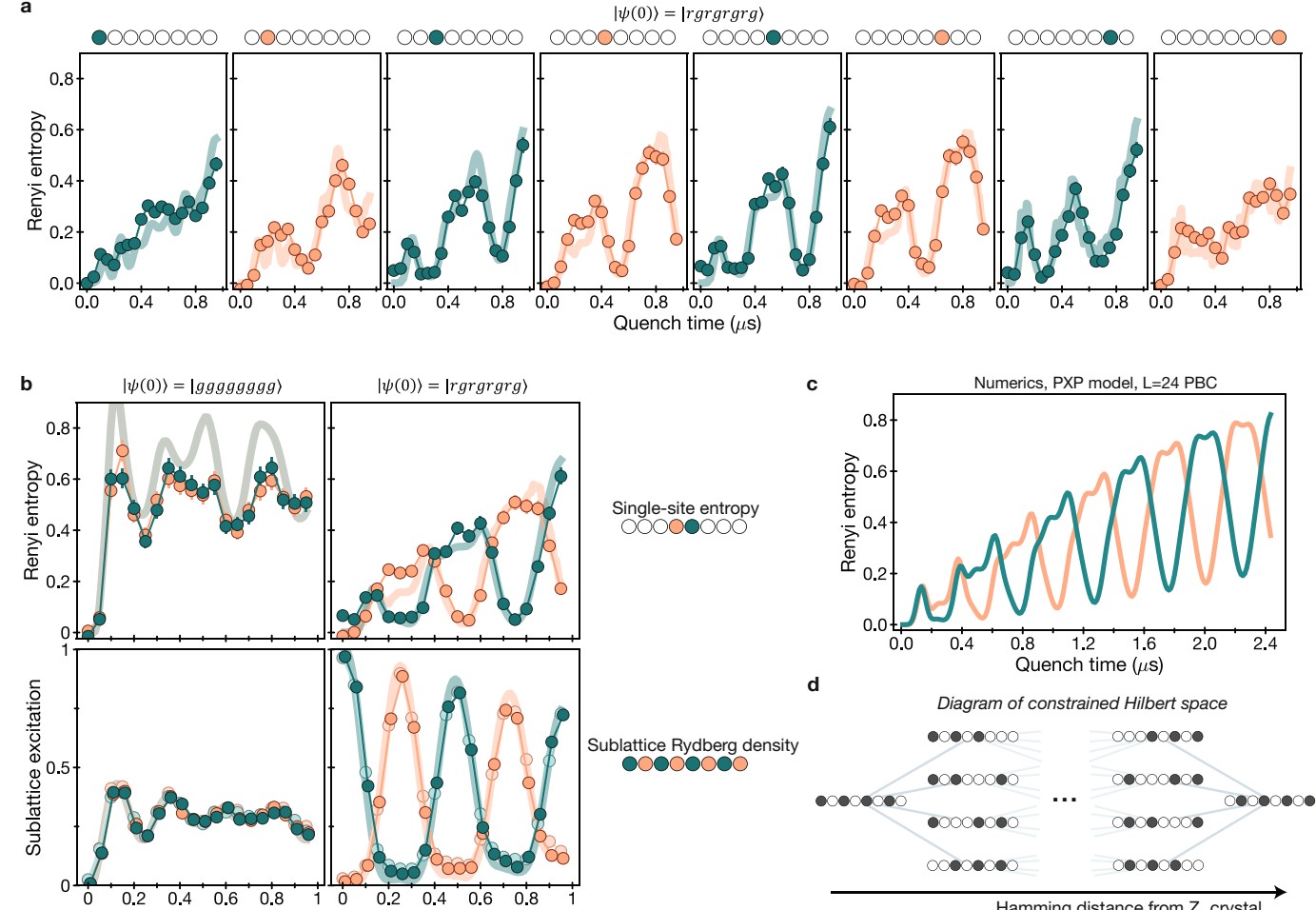

**Extended Data Fig. 10 | Local observables and entanglement entropy for quantum many-body scars. a**, Experimentally measured single-site entropy for each site in the 8-atom chain when quenching from the scarred $|\mathbb{Z}_2\rangle$ state, including the classical entropy subtraction. Solid curves plot exact, ideal (imperfection-free) numerics of $H_{\mathrm{Ryd}}$ (Eq. 3); excellent agreement between data and numerics is found for every atom in the chain. **b**, (Top) Same data as Fig. 4f of the main text, showing single-site entropy of the middle two atoms in the chain, for two different initial states[75]. (Bottom) Measurements of the many-body state in the Z-basis with the interferometry circuit turned off. Characteristic of the scars from the $|\mathbb{Z}_2\rangle = |rgrgrgrg\rangle$ state, the Rydberg excitation probability on the sublattices exhibits periodic oscillations[71]. In the bottom row, the dark data points are measured in the $\{|1\rangle, |r\rangle\}$ basis, and the faint data points are measured in the $\{|0\rangle, |1\rangle\}$ basis after the coherent mapping sequence. Measurements in both bases agree well with exact numerics (solid lines), which we emphasize has no free fit parameters and does not account for any experimental imperfections, such as detection infidelity. Moreover, the data indicate the high fidelity of preparation into the $|\mathbb{Z}_2\rangle$ state by use of local

Rydberg $\pi$ pulses. In plotting, we delay the theory curves and the $\{|1\rangle, |r\rangle\}$ basis measurement by 10 ns to account for the fact that the Raman $\pi$ pulse we apply cuts off the final 10 ns of the Rydberg evolution, when measuring in the $\{|0\rangle, |1\rangle\}$ basis. **c**, Numerical simulations of the single-site Renyi entropy on two adjacent sites in the idealized 'PXP' model of perfect nearest-neighbor blockade[14]. The system size is 24 atoms with periodic boundary conditions, showing the same out-of-phase oscillations in the entanglement entropy of the two sublattices. **d**, Diagram of the constrained Hilbert space of the system[14]. The early-time, out-of-phase entropy oscillations[75] of the scars can be understood in this constrained Hilbert space picture, where the scar dynamics are known to take the state from the left end ($|rgrgrgrg\rangle$) to the right end ($|grgrgrgr\rangle$) (dark circles represent $|r\rangle$ and white circles represent $|g\rangle$)[14]. Near these crystalline ends of this constrained Hilbert space, the Rydberg atoms can fluctuate (high entropy), but the ground state atoms are pinned (low entropy). Our analysis shows that entanglement between atoms on the same sublattice contributes to the eventual degradation of the revival fidelity of the $|\mathbb{Z}_2\rangle$ state.