## [Peer Review File · Nature]

Manuscript Title: A quantum processor based on coherent transport of entangled atom arrays

Reviewer Comments & Author Rebuttals

Reviewer Reports on the Initial Version:

Referees' comments:

Referee #1 (Remarks to the Author):

The manuscript studies a platform for quantum computation based on Rydberg atoms trapped by individual optical tweezers. Within this framework, the authors have previously demonstrated the ability to perform high fidelity quantum gates between nearest neighbor qubits. The big breakthrough in the present manuscript is the demonstration that a subset of atoms can be transported during the runtime of the algorithm and faster than the decoherence time for a single qubit. The authors use this property to apply quantum gates on a subset of atoms. This allows the authors to change the connectivity of the qubits while the quantum algorithm is performed, and therefore, apply quantum gates between qubits, which are initially not nearest neighbors. In the referee's opinion, this is a true milestone achievement and demonstrates that quantum computers based in Rydberg atoms possess properties, which are superior to any other quantum computing platform so far available. The authors use this property for application of several proof of principle experiments. Especially, the authors prepare several highly entangled states, which have important applications in error correction schemes for quantum computers such as the 7 qubit Steane code, a surface code with 13 qubits and the toric code on a torus with 16 qubits. Furthermore, they also demonstrate the ability of their platform to work as a hybrid system for quantum simulations, i.e., combining coherent time evolution within a quantum simulation with quantum gates of a digital quantum computer for probing the quantum state. Especially, the authors probe the Renyi entropy within many-body scars.

The manuscript is well written, but there are several aspects, which should be clarified.

(1) The most important one is the claim in the abstract that the authors realize a surface code with 19 qubits and toric code with 24 qubits. However, the surface code state is only realized for 13 qubits, while the other 6 qubits act as ancilla. The correct statement would be, that the authors prepare a surface code with 13 qubits implemented in a setup with a total of 19 qubits. The same misleading statement happens for the toric code: the state of the toric code involves 16 qubits and 8 qubits act as ancilla. Again, the state is only realized within a setup with a total of 24 qubits, and involves only 16 qubits.

(2) The authors very poorly describe the procedure for probing the entanglement. Especially, the sentence "... we apply an additional $\pi/2$ pulse with a variable phase that results..." does not make

sense, and also the method section does not clarify the issue. After carefully reading Ref.[5] the referee could guess, that the authors mean that they apply an additional single qubit phase gate $Z[\phi]$ before the final $\pi/2$ pulse. If correct, this part should be properly explained.

(3) The authors mention that all stabilizer QEC states are equivalent to graph states up to single qubit Clifford rotations. While this statement is certainly correct, it is also misleading in the present context: an important aspect of QEC states is that all stabilizers are local as non-local errors can not be corrected. However, the corresponding graph states always include highly non-local connections. Therefore, the implementation of the graph state instead of the stabilizer state can not serve for quantum error correction in codes involving many qubits. This aspect is not yet important for the 7-qubit Steane code, but can no longer be ignored for the toric code and surface code. This part of the manuscript should be properly rewritten. Especially, as the authors do not use this equivalence for the realization of the toric code.

(4) It is not clear to the referee how exactly the surface code state and the toric code states are prepared. The authors mention that they perform a projective measurement of the ancilla qubits. However, the measurement values will be random in each shot of the experiment, but the desired topological state will only be obtained if the measurement results of the ancillas is 1. Do the authors perform post selection, that they account only for the events with this outcome? This would fundamentally weaken the message of the manuscript as this procedure is impractical for large system. If the authors do not rely on such a post selection, they should explain how the preparation of the correct state is achieved. Or do the authors in each shot of the experiment create a state with a random number of anyonic excitation and account for the excitations for the evaluation for the stabilizer states? This approach would be very disappointing as then the authors do not create the toric code state but rather a 24 qubit graph state. This important aspects has to be clarified and properly explained in the manuscript.

In summary, the authors present a true breakthrough by the first demonstration of transport for qubits changing the connectivity during a quantum algorithm with several important and interesting applications. These results certainly deserve a publication in Nature. However, there are some aspects on the different applications, which should be clarified before the manuscript can be accepted for publication.

Referee #2 (Remarks to the Author):

For several years now, there has been an ongoing effort worldwide to make neutral atoms in arrays of optical tweezers a viable platform for quantum computation. Till today, different aspects of this platform have been demonstrated, including initialization and rearrangement of tweezers with real-time feedback, shuttling of qubits in a two-dimensional plane, and parallel application of two-qubit gates. Many of these achievements have been made by the group of Prof. Lukin. The manuscript by Bluvstein et al. reports on a significant step forward - generation of complex quantum circuits and many-qubit entangled states through a successive application of these steps. The authors demonstrate the generation of topological quantum states incorporating up to 24 atoms, with the prospect of leading to fault-tolerant quantum computation in the future. In addition, they show how

they can measure elusive observables in quantum many-body dynamics, such as the entanglement entropy. I suspect that it should also be possible to identify order parameters hidden in non-local correlation functions with a similar approach. Specifically, they demonstrate both area and volume laws entanglement growth and period oscillations in so-called quantum scars states. These remarkable achievements set a new milestone in the route towards quantum computation, namely the demonstration of the first "atomic quantum processor" based on neutral atoms in reconfigurable optical tweezers. For this reason, I believe this paper fits Nature.

That being said, two questions are not adequately addressed. First, what is this platform's computational power, and how does it compare with other platforms? Second, how scalable is the current approach? Regarding the computational power, I expected the authors to characterize their processor with either randomized benchmarking (PRL 106, 180504) or quantum volume (PRA100, 032328), similar, for example, to Ref 24. Instead, the authors provide measurements of stabilizers, but it is not clear how to relate these numbers to the other measures of computational power or state fidelity. For example, given the generation fidelity of the 12-atom cluster state, can it be used for a measurement-based computation, and if yes, what is the number of logical qubits that this state can represent? Similar questions can be asked about the Steane, surface, and toric codes.

The question of scalability is, of course, related to the infidelity of gate operations and qubit shuttling. While these topics are discussed in the Methods section, they are absent from the main text. In the Methods, the authors analyze the sources of infidelities. They comment on ways to reduce them and then use numerical simulation to project their ability to reach fault tolerance in the future. I think a short version of this discussion must appear in the paper. Since the central point of this paper is showcasing the power of the atomic processor, a demonstration of quantum circuits and generation of graph states must be complemented with a discussion of how the fidelity scales with the number of layers, qubits, and gate or shuttling operations.

One of the key aspects of the atomic processor is the fast shuttling of atoms in tweezers. To mitigate decoherence during the movement, the authors apply dynamical decoupling. The authors perform a parity measurement on pairs of atoms in a Bell state and demonstrate that the entanglement survives if the average velocity during the movement is not too high. It is worth mentioning that the question of how fast an atom can be moved between two distant locations is connected to the notion of quantum speed limits. See, for example, the recent paper by Lam et al. in Phys. Rev. X 11, 011035 (2021). Given the physical resources (e.g., trap depth), quantum speed limits set an upper bound to the maximum rate of operation for the atomic processor. It is worth considering and mentioning these fundamental limits in a paper describing a processor that relies so heavily on these movements to change the connectivity of the quantum circuit.

Referee #3 (Remarks to the Author):

In this paper the authors exploit the high versatility of reconfigurable tweezer arrays of rubidium atoms to engineer non-local connectivity demonstrating a novel quantum processor. They explore various systems: graph states (1D and 2D), surface code and toric code states. Additionally, they apply the same technique to investigate quantum many-body scars via direct measurement of the

Renyi entanglement entropy following a quench. This work represents a large step forwards for the Rydberg tweezer platform. With the work on quantum circuits the authors have demonstrated the versatility and capability of their platform, and in exploring the many-body scars they have confirmed previously theoretical predictions for the entanglement entropy of quantum scars. The results presented in this paper are both novel, and broad. However, the authors have presented it in a clear and logical way, starting from the conceptual simple experiment of entangling neighbouring atoms, transporting them and measuring the entanglement and the Bell state fidelity with and without movement. The methods section shows how residual errors caused by e.g., SPAM, Rydberg lifetime, and technical limitations (power stability of beams) etc. contribute to the observed results in a thorough way.

They then move on to discuss programmable circuits and graph states, where they drive parallel CZ gates between two atoms to mimic links on the graph state. I found the methodology for the preparation of the different states well explained. The authors declare error corrected values for $\langle Z_L \rangle$ and $\langle X_L \rangle$ which confirm the preparation of the logical qubit state. Following this they extend the platform to use transportable ancillary qubits to drive operations between distant atoms, in these experiments the data qubits only ever interact with the ancillary qubits. They demonstrate preparation of the surface code and toric code states, with high fidelity, especially when correcting for known experimental errors. The raw values seem quite low, however in the outlook the authors discuss how to decrease the errors, and also to implement mid-circuit readout. My understanding is that this would allow one to prepare the logical state with the stated error, measure the state using the ancillary qubits, and then continue the experiment. This would then remove the preparation error as one could neglect runs with non-perfect preparation, am I correct here? Not being an expert in quantum circuits, I have a question for the authors regarding the toric code state. What are the seven plaquettes/stars? From the diagram I would expect eight.

The final results presented by the authors are exploring applications to quantum simulation. They quench the system and observe the growth of entanglement energy by measuring the Renyi entanglement entropy. They do this for two systems, the first has all atoms prepared in the ground state $|g\rangle$, the second is the anti-ferromagnetic ground state $|grgr\dots\rangle$. They observe that for the second system, the rate of the entropy growth is suppressed. They even find that the two sublattices (one containing state $|g\rangle$, and the other $|r\rangle$), are disentangled from one another.

In the outlook the authors clearly identify areas for improvement on their reported results.

However, I would question the claim that the improvements stated would allow 'for direct scaling to deep quantum circuits operating on thousands of neutral atom qubits' [line 223]. On this scale the system would surely be limited by the 10s vacuum lifetime, which for 1000 atoms would give an ensemble lifetime of $10/1000 = 10$ ms. I would like this sentence to be clarified.

With these results the authors have demonstrated applications of their technique, which is unique to reconfigurable arrays of qubits, within both the fields of quantum computation and simulation of many-body physics. The work on the toric code is particularly exciting, as it demonstrates how one could use this platform to explore 3D, or periodic boundary systems, on a 2D array. Their conclusions are well supported, and the data has been carefully taken such that all aspects of the experimental setup and the findings are understood.

I think this paper is exciting for the quantum and AMO communities, as well as of general interest for Nature readers. I would therefore recommend publication of this article, and commend the authors on their impressive results.

Referee responses and summary of revisions

Nature Manuscript number: 2021-12-19162 Bluvstein

January 6, 2022

We would like to thank all Referees for their careful reading of our manuscript and many useful comments and suggestions. In what follows we address all comments point by point and indicate revisions when appropriate. When addressing the referees' comments with changes to the text, we have tried to add as few words as possible (and removed words where applicable), as per the editor's request. In totality we have added a net +13 words to the main text.

Reviewer: 1

The manuscript studies a platform for quantum computation based on Rydberg atoms trapped by individual optical tweezers. Within this framework, the authors have previously demonstrated the ability to perform high fidelity quantum gates between nearest neighbor qubits. The big breakthrough in the present manuscript is the demonstration that a subset of atoms can be transported during the runtime of the algorithm and faster than the decoherence time for a single qubit. The authors use this property to apply quantum gates on a subset of atoms. This allows the authors to change the connectivity of the qubits while the quantum algorithm is performed, and therefore, apply quantum gates between qubits, which are initially not nearest neighbors. In the referee's opinion, this is a true milestone achievement and demonstrates that quantum computers based in Rydberg atoms possess properties, which are superior to any other quantum computing platform so far available. The authors use this property for application of several proof of principle experiments. Especially, the authors prepare several highly entangled states, which have important applications in error correction schemes for quantum computers such as the 7 qubit Steane code, a surface code with 13 qubits and the toric code on a torus with 16 qubits. Furthermore, they also demonstrate the ability of their platform to work as a hybrid system for quantum simulations, i.e., combining coherent time evolution within a quantum simulation with quantum gates of a digital quantum computer for probing the quantum state. Especially, the authors probe the Renyi entropy within many-body scars.

We thank the referee for their positive evaluation of our manuscript.

The manuscript is well written, but there are several aspects, which should be clarified.

(1) The most important one is the claim in the abstract that the authors realize a surface code with 19 qubits and toric code with 24 qubits. However, the the surface code states is only realized for 13 qubits, while the other 6 qubits act as ancilla. The correct statement would be, that the authors prepare a surface code with 13 qubits implemented in a setup with a total of 19 qubits. The same misleading statement happens for the toric code: the state of the toric code involves 16 qubits and 8 qubits act as ancilla. Again, the state is only realized within a setup with a total of 24 qubits, and involves only 16 qubits.

We thank the referee for highlighting this concern and possible misinterpretation. As proposed by the referee, we have changed the abstract to read (underline denotes change)

“Furthermore, we shuttle entangled ancilla arrays to realize a surface code state with 13 data and 6 ancillary qubits and a toric code state on a torus with 16 data and 8 ancillary qubits.”

(2) The authors very poorly describe the procedure for probing the entanglement. Especially, the sentence “..., we apply an additional $\pi/2$ pulse with a variable phase that results...” does not make sense, and also the method section does not clarify the issue. After carefully reading Ref.[5] the referee could guess, that the authors mean that they apply an additional single qubit phase gate $Z[\phi]$ before the final $\pi/2$ pulse. If correct, this part should be properly explained.

We agree. In response we have modified the sentence in the main text to read

To measure the resulting entangled-state fidelity, we apply a variable single-qubit phase gate before a final $\pi/2$ pulse, resulting in oscillations of the two-atom parity $\langle \sigma_1^z \sigma_2^z \rangle$.

(3) The authors mention that all stabilizer QEC states are equivalent to graph states up to single qubit Clifford rotations. While this statement is certainly correct, it is also misleading in the present context: an important aspect of QEC states is that all stabilizers are local as non-local errors can not be corrected. However, the corresponding graph states always include highly non-local connections. Therefore, the implementation of the graph state instead of the stabilizer state can not serve for quantum error correction in codes involving many qubits. This aspect is not yet important for the 7-qubit Steane code, but can no longer be ignored for the toric code and surface code. This part of the

manuscript should be properly rewritten. Especially, as the authors do not use this equivalence for the realization of the toric code.

We agree that in general the connection between QEC states and graph states can be complex. Since this comment is not central to our experiments and does not affect any conclusions, we eliminated it from the manuscript:

~~An important class of graph states are quantum error correcting (QEC) codes, where the graph state stabilizers manifest as the stabilizers of the QEC code and can be measured to correct errors on an encoded logical qubit. All stabilizer QEC states are equivalent to some graph state up to single-qubit Clifford rotations, hence the ability to generate arbitrary graph states allows one to readily prepare a wide variety of QEC states.~~

(4) It is not clear to the referee how exactly the surface code state and the toric code states are prepared. The authors mention that they perform a projective measurement of the ancilla qubits. However, the measurement values will be random in each shot of the experiment, but the desired topological state will only be obtained if the measurement results of the ancillas is 1. Do the authors perform post selection, that they account only for the events with this outcome? This would fundamentally weaken the message of the manuscript as this procedure is impractical for large system. If the authors do not rely on such a post selection, they should explain how the preparation of the correct state is achieved. Or do the authors in each shot of the experiment create a state with a random number of anyonic excitation and account for the excitations for the evaluation for the stabilizer states? This approach would be very disappointing as then the authors do not create the toric code state but rather a 24 qubit graph state. This important aspects has to be clarified and properly explained in the manuscript.

We thank the Referee for this comment. Remarkably, the ancilla technique we used does not require any postselection, and generates a topological state useful for error correction regardless of the measured ancilla values (as long as they are known). More specifically, the stabilizer results we plot as the “Z stars” are precisely those shown in the Figure 3a schematic: X on the ancilla qubit and $ZZZZ$ on the surrounding data qubits. This is a stabilizer of the underlying graph state and so requires no postselection: the measured value would always be +1 if there were no experimental imperfections. One can also reinterpret this quantity as the stabilizer product $ZZZZ$, with a redefined correct stabilizer value of ± 1 given by the ancilla measurement of ± 1 . This aspect has been pointed out in early theoretical studies on graph states, see e.g. [Raussendorf PRL 2007 and Raussendorf NJP 2007] and [Briegel Nat Phys 2009]. The ability

of this technique to generate a useful topological state independent of the ancilla measurement values is commented on explicitly in e.g. [Bolt PRL 2016] and [arXiv:2112.01519].

This remarkable result can be understood in several ways. First, the ancilla values merely amount to a redefinition of the Z star stabilizers, and one can simply redefine the topological state with the appropriate $S_i \rightarrow -S_i$. In order to arrive at the “standard” stabilizer definition, one could imagine experimentally performing mid-circuit readout on the ancillas and then applying an appropriate X string to pair “anyons” at every ancilla measuring $X = -1$. However, it is known that deterministic single-qubit rotations cannot change the topological nature of a state, and so this procedure cannot change the topological properties of the created state. One could argue that single-qubit rotations allow one to measure the topological state in the appropriate basis, but this X string simply flips the definition of $|0\rangle \leftrightarrow |1\rangle$ of each affected qubit, which is a trivial basis change that can simply be performed in post-processing (as is effectively done in our work here).

We emphasize that this technique also does not impose any limitations on practical QEC operation, even without any active, real-time correction on physical qubits. E.g. it is known for surface code operation that, following each stabilizer measurement round, corrections can be implemented exclusively in the classical control software, as discussed in Fowler PRA 2012 [Ref 38]. The correction strings for pairing anyons are applied in software, simply redefining the affected stabilizers; and if the correction string crosses the X_L or Z_L operator then the logical qubit is corrected by applying Z_L or X_L , which is *also* applied in the classical control software and is properly commuted through when applying S or T gates. The ancilla technique here can be understood in the same fashion, with all corrections applied in software. To address this important point, we have added the following sentence:

The graph state stabilizers now transform into the X -plaquettes, the Z -stars (with value ± 1 for a measurement outcome of ± 1 of the central ancilla), and the logical X_L operator. Remarkably, this procedure creates a topologically ordered state in a constant-depth circuit [Bravyi2006, Raussendorf2007], where measured ancilla values can be used for redefining stabilizers, which can be handled in software for practical QEC operation [Fowler2012].

In summary, the authors present a true breakthrough by the first demonstration of transport for qubits changing the connectivity during a quantum algorithm with several important and interesting applications. These results certainly deserve a publication in Nature. However, there are some aspects on the different applications, which should be clarified before the manuscript can be accepted for publication.

We thank the referee again for their positive evaluation of our manuscript and helpful comments.

Reviewer: 2

For several years now, there has been an ongoing effort worldwide to make neutral atoms in arrays of optical tweezers a viable platform for quantum computation. Till today, different aspects of this platform have been demonstrated, including initialization and rearrangement of tweezers with real-time feedback, shuttling of qubits in a two-dimensional plane, and parallel application of two-qubit gates. Many of these achievements have been made by the group of Prof. Lukin. The manuscript by Bluvstein et al. reports on a significant step forward - generation of complex quantum circuits and many-qubit entangled states through a successive application of these steps. The authors demonstrate the generation of topological quantum states incorporating up to 24 atoms, with the prospect of leading to fault-tolerant quantum computation in the future. In addition, they show how they can measure elusive observables in quantum many-body dynamics, such as the entanglement entropy. I suspect that it should also be possible to identify order parameters hidden in non-local correlation functions with a similar approach. Specifically, they demonstrate both area and volume laws entanglement growth and period oscillations in so-called quantum scars states. These remarkable achievements set a new milestone in the route towards quantum computation, namely the demonstration of the first "atomic quantum processor" based on neutral atoms in reconfigurable optical tweezers. For this reason, I believe this paper fits Nature.

We thank the referee for their positive evaluation of our manuscript. Identifying hidden order parameters using the techniques here would indeed be an exciting future direction.

That being said, two questions are not adequately addressed. First, what is this platform's computational power, and how does it compare with other platforms? Second, how scalable is the current approach? Regarding the computational power, I expected the authors to characterize their processor with either randomized benchmarking (PRL 106, 180504) or quantum volume (PRA100, 032328), similar, for example, to Ref 24. Instead, the authors provide measurements of stabilizers, but It is not clear how to relate these numbers to the other measures of computational power or state fidelity. For example, given the generation fidelity of the 12-atom cluster state, can it be used for a measurement-based computation, and if yes, what is the number of logical qubits that this state can represent? Similar questions can be asked about the Steane, surface, and toric codes.

The question of scalability is, of course, related to the infidelity of gate operations and qubit shuttling. While these topics are discussed in the Methods section, they are absent from the main text. In the Methods, the authors analyze the sources of infidelities. They comment on ways to reduce them and then use numerical simulation to project their ability to reach fault tolerance in the future. I think a short version of this discussion must appear in the paper. Since

the central point of this paper is showcasing the power of the atomic processor, a demonstration of quantum circuits and generation of graph states must be complemented with a discussion of how the fidelity scales with the number of layers, qubits, and gate or shuttling operations.

We thank the referee for these questions and suggestions. The question of computational power, and its comparison to other systems, is highly dependent on the particular application. In the current work we focus primarily on creating different states with parallel quantum circuits. To access the “computational power” of these states and comparison to other systems, one can discuss the logical qubit fidelities as well as their proximity to the code threshold. To address this point, we have added the following to the caption of Extended Data Figure 7, which overviews our logical qubit fidelities:

We remark that the observed fidelities are comparable to similar demonstrations in state-of-the-art experiments with other platforms [Satzinger2021] [Erhard2021]. These will need to be improved to surpass the threshold for practical error correction [Fowler 2012] (see Methods text).

At the same time, there is a sense in which our system exhibits a significant advantage in the computational power for logical qubit processing, due to the ability to realize nonlocal connectivity and interlace arrays of qubits *in a highly parallel fashion*. This is commented on in the outlook: **Furthermore, the ability to reconfigure and interlace our arrays will allow efficient, parallel execution of transversal entangling gates between many logical qubits [Wang2003, Fowler2012]. In addition, these techniques also enable implementation of higher-dimensional or nonlocal error correcting codes with more favorable properties [Bombin2015a, Breuckmann2021].** We believe that this discussion, as well as the comparison to other platforms added above, aptly illustrates the computational power of our system with respect to logical qubit computation within the scope of the current work.

For the 12-atom 1D cluster state and its use for measurement-based quantum computation, the per-qubit fidelity would translate into the probability that a gate in measurement-based quantum computation is properly executed. For the 1D cluster state in a line, such a measurement-based quantum computation would correspond to performing rotations on a single qubit with an error rate of a few percent, since the average stabilizers $\langle S_i \rangle = 0.91$ if accounting for SPAM, corresponding to an error probability of approximately a few percent per-qubit since there are predominantly three qubits per stabilizer and stabilizers are either -1 or +1 ($\langle S_i \rangle = 2 * (\text{probability of no error}) - 1$). We have added this discussion to the text as suggested by the referee:

Across all twelve stabilizers we find an average $\langle S_i \rangle = 0.87(1)$ (Fig. 2c)

(accounting for state-preparation-and-measurement SPAM errors would yield $\langle S_i \rangle = 0.91(1)$), certifying biseparable entanglement in a cluster state (all $\langle S_i \rangle > 0.5$ [Toth 2005]). The measured fidelities would correspond to a few percent error-per-operation for a measurement-based quantum computation [Raussendorf2001 Raussendorf2007]

To address the question of scalability and discussion of error sources and their dependence on qubit number, layer number, shuttling, etc, we have added the following to the main text at the end of the topological state section:

Our measured fidelities are in good agreement with numerical simulations of the circuit (Extended Data Fig. 6), wherein each qubit experiences a per-layer error rate independent of the number of qubits or the shuttling process, indicating that errors in CZ gates (fidelity $\approx 97.5\%$, Methods [Levine 2019]) constitute our dominant error source.

The high fidelity of the motion is shown in Figure 1, and per the referee's suggestion we have also added a mention in the main text to the high fidelity of the single-qubit operations:

We store quantum information in magnetically insensitive clock states within the ground state hyperfine manifold of ^{87}Rb atoms [Beugnon2007], and implement robust single-qubit Raman rotations (scattering error per π -pulse $\sim 7 \times 10^{-5}$) [Levine2021], realized by composite pulses that are robust to pulse errors (Extended Data Fig. 3) [Vandersypen2005].

Our approach is indeed scalable, since atoms trapped in optical tweezers are completely independent from each other; moreover the shuttling is observed to have a negligible effect on errors (Fig. 1). The main barrier to larger qubit numbers in the present work is the lack of local Rydberg addressing, which is discussed in the outlook, and nicely complements the addition above suggested by the referee: **Local Rydberg excitation on subsets of qubit pairs would eliminate residual interactions from unintended atoms, allowing parallel, independent operations on arrays with significantly higher qubit densities. Two-qubit gate fidelity can be improved using higher Rydberg laser power or more efficient delivery methods, as well as more advanced atom cooling., and shortly followed by ...with projected fidelity improvements theoretically surpassing the surface code threshold (Methods)., in reference to the aforementioned improvements to two-qubit gate fidelity. With local addressing, the qubit number could be increased significantly, to levels discussed earlier in the paper: We note that the entanglement transport in Figure 1b corresponds to moving quantum information across a region of space that can in principle host ~ 2000 qubits (at an atom separation of $3 \mu\text{m}$), on a timescale corresponding to $< 10^{-3} T_2$ (Extended Data Fig. 3), directly enabling applications**

in large-scale quantum information systems. With the above additions as suggested by the referee, now complementing these three already-existing comments, we believe this constitutes an apt discussion of the scalability of the platform.

One of the key aspects of the atomic processor is the fast shuttling of atoms in tweezers. To mitigate decoherence during the movement, the authors apply dynamical decoupling. The authors perform a parity measurement on pairs of atoms in a Bell state and demonstrate that the entanglement survives if the average velocity during the movement is not too high. It is worth mentioning that the question of how fast an atom can be moved between two distant locations is connected to the notion of quantum speed limits. See, for example, the recent paper by Lam et al. in Phys. Rev. X 11, 011035 (2021). Given the physical resources (e.g., trap depth), quantum speed limits set an upper bound to the maximum rate of operation for the atomic processor. It is worth considering and mentioning these fundamental limits in a paper describing a processor that relies so heavily on these movements to change the connectivity of the quantum circuit.

We thank the referee for these references. In the main text we have added:

Performing this experiment as a function of movement speed [Lam2021] shows that fidelity remains unchanged until the total separation speed becomes $> 0.55 \mu\text{m}/\mu\text{s}$, corresponding to the onset of atom loss (Fig. 1d).

And in our Methods section “Movement effects on atom heating and loss,” where we discuss in greater quantitative detail our movement limitations, we have added:

Move speed could be further improved with different $a(t)$ profiles, but inevitably with finite resources such as trap depth, quantum speed limits will eventually prevent arbitrarily fast motion of qubits across the array [Lam2021].

Reviewer: 3

In this paper the authors exploit the high versatility of reconfigurable tweezer arrays of rubidium atoms to engineer non-local connectivity demonstrating a novel quantum processor. They explore various systems: graph states (1D and 2D), surface code and toric code states. Additionally, they apply the same technique to investigate quantum many-body scars via direct measurement of the Renyi entanglement entropy following a quench. This work represents a large step forwards for the Rydberg tweezer platform. With the work on quantum circuits the authors have demonstrated the versatility and capability of their platform, and in exploring the many-body scars they have confirmed previously theoretical predictions for the entanglement entropy of quantum scars. The results presented in this paper are both novel, and broad. However, the authors have presented it in a clear and logical way, starting from the conceptual simple experiment of entangling neighbouring atoms, transporting them and measuring the entanglement and the Bell state fidelity with and without movement. The methods section shows how residual errors caused by e.g., SPAM, Rydberg lifetime, and technical limitations (power stability of beams) etc. contribute to the observed results in a thorough way. They then move on to discuss programmable circuits and graph states, where they drive parallel CZ gates between two atoms to mimic links on the graph state. I found the methodology for the preparation of the different states well explained. The authors declare error corrected values for Z_L and X_L which confirm the preparation of the logical qubit state. Following this they extend the platform to use transportable ancillary qubits to drive operations between distant atoms, in these experiments the data qubits only ever interact with the ancillary qubits.

We thank the referee for their positive evaluation of our manuscript.

They demonstrate preparation of the surface code and toric code states, with high fidelity, especially when correcting for known experimental errors.

We emphasize that all plotted stabilizers and logical operators are directly extracted from the raw bitstring data measured experimentally, without any additional correction for experimental errors. The data titled “raw” consists of plotting the measured logical operator expectation values over all instances, and the data titled “error detection” consists of plotting the measured logical operator expectation values only over the instances where none of the stabilizers detect an error (as opposed to separately accounting for experimental errors). While this is stated clearly for the Steane code, to help clarify this point for the surface and toric code as well, we have added the following to the main text:

We find a raw value of $\langle X_L \rangle = 0.64(3)$, and a corrected value of $\langle \bar{X}_L \rangle = 1_{-0.01}^{+0}$ using the measured stabilizers for error detection (with

35(1)% probability of no detected errors), demonstrating preparation of this topological QEC state (see also Extended Data Fig. 7).

We do note however that indeed as seen in ED Fig. 6, our known experimental errors do quantitatively explain the graph state fidelities.

The raw values seem quite low, however in the outlook the authors discuss how to decrease the errors, and also to implement mid-circuit readout.

While we agree that the raw logical qubit error rates will indeed need to be improved in the ways discussed, it is comparable to state-of-the-art results in other platforms; e.g. our distance-3 surface code error rates plotted in ED Fig. 7 are within a factor of 2 of the distance-3 surface code in Fig. 4D in Satzinger et al Science 2021 (Google superconducting qubits). To emphasize this point (see also Referee 2's comments), we have added the following to the caption of ED Fig. 7:

We remark that the observed fidelities are comparable to similar demonstrations in state-of-the-art experiments with other platforms [Satzinger2021] [Erhard2021]. These will need to be improved to surpass the threshold for practical error correction [Fowler 2012] (see Methods text).

My understanding is that this would allow one to prepare the logical state with the stated error, measure the state using the ancillary qubits, and then continue the experiment. This would then remove the preparation error as one could neglect runs with non-perfect preparation, am I correct here?

Indeed, the referee is describing the process for doing practical error-detection-assisted state preparation using ancillary qubits and mid-circuit readout. One can prepare the state, measure stabilizers and/or the logical qubit state with ancillas, and only proceed if no errors are detected which would greatly increase preparation fidelity (to a limit affected by the ancilla-based measurement fidelity).

Not being an expert in quantum circuits, I have a question for the authors regarding the toric code state. What are the seven plaquettes/stars? From the diagram I would expect eight.

Although there are eight plaquettes and stars, there are in fact only seven *independent* plaquettes and stars. In a QEC code, the set of stabilizers should be independent such that none of the stabilizers can be written as the product of

the others. Due to the periodic boundary conditions of the torus, the standard approach to have an independent set of stabilizers involves simply removing one of the eight stabilizers from the set (which stabilizer is irrelevant, as by definition that stabilizer can be written as the product of the others). Another way to understand this is by counting degrees of freedom: the surface code has 13 data qubits, comprising 12 stabilizers and 1 logical qubit, and the toric code has 16 data qubits, comprising 14 stabilizers and 2 logical qubits. This is discussed in Kitaev's original paper (the cited reference 9) and standard toric code reviews and literature. To clarify this point, we have added the following to the main text:

The state we prepare has seven (due to periodic boundary conditions) independent X -plaquettes and seven independent Z -stars. Moreover, due to the topological properties of this graph, two independent logical qubits can be encoded with logical operators $X_L^{(1)}, Z_L^{(1)}$ and $X_L^{(2)}, Z_L^{(2)}$ that wrap around the entire torus along two topologically distinct directions [Kitaev 2003].

The final results presented by the authors are exploring applications to quantum simulation. They quench the system and observe the growth of entanglement energy by measuring the Renyi entanglement entropy. They do this for two systems, the first has all atoms prepared in the ground state g , the second is the anti-ferromagnetic ground state $grgr$. They observe that for the second system, the rate of the entropy growth is suppressed. They even find that the two sublattices (one containing state g , and the other r , are disentangled from one another.

In the outlook the authors clearly identify areas for improvement on their reported results. However, I would question the claim that the improvements stated would allow 'for direct scaling to deep quantum circuits operating on thousands of neutral atom qubits' [line 223]. On this scale the system would surely be limited by the 10s vacuum lifetime, which for 1000 atoms would give an ensemble lifetime of $10/1000 = 10$ ms. I would like this sentence to be clarified.

We thank the referee for this request. Although the relevant quantity depends on the specific quantum circuit application, for quantum error correction with large surface code arrays, the relevant quantity is the per-qubit error rate, and whether that error rate is above or below the code threshold error rate ($\sim 1\%$ for surface code). Indeed, it is the remarkable fact of quantum error correction that once the per-qubit error rate is below the threshold, *increasing* the number of qubits in a logical qubit results in a *decreasing* error rate on the logical qubit. In such a case, one should focus on the per-qubit error rate and not the ensemble error rate.

The rate of atom loss to vacuum is an additional source of per-qubit error, that is sometimes referred to as an erasure error. Much like X and Z errors, it

can be handled in our system in a hardware-efficient manner with a comparable overhead using techniques such as those discussed in [Cong et al 2021]. Accordingly, in order to preserve a logical qubit in the face of decoherence and atom loss errors, one requires that the per-qubit error rate is $< 1\%$ for each stabilizer measurement cycle, which would take ~ 1 ms as discussed in the text. During the 1 ms QEC round, the rate of atom loss due to the 10s vacuum lifetime is roughly 0.01% per-qubit, which is a significantly smaller error rate than the $\sim 1\%$ threshold. As such, although atom loss needs to be properly accounted for (and the atom properly replaced using a reservoir of extra atoms), from the perspective of error correction the rate of loss due to vacuum is currently negligible. In principle, to run a quantum circuit for ~ 1 s (corresponding to a very large depth of ~ 10000 operations per qubit) would require having an atom reservoir of size 10% of the data qubit array in order to continuously replace data qubits which are identified as being lost. In an even more sophisticated approach, the circuit depth can be extended indefinitely by constantly reloading atoms throughout the circuit from e.g. a MOT.

In the Methods section “Qubit coherence and dynamical decoupling,” we have added the following to address this comment and discuss the correctability of atom loss:

For practical QEC operation, atom loss can be detected in a hardware-efficient manner [Cong2021] and the atom then replaced from a reservoir, which could in principle be continuously reloaded by a MOT for reaching arbitrarily deep circuits.

Furthermore, we adjusted the sentence in the main text by removing “direct” to read:

These technical improvements should allow for ~~direct~~ scaling to deep quantum circuits operating on thousands of neutral atom qubits.

With these results the authors have demonstrated applications of their technique, which is unique to reconfigurable arrays of qubits, within both the fields of quantum computation and simulation of many-body physics. The work on the toric code is particularly exciting, as it demonstrates how one could use this platform to explore 3D, or periodic boundary systems, on a 2D array. Their conclusions are well supported, and the data has been carefully taken such that all aspects of the experimental setup and the findings are understood. I think this paper is exciting for the quantum and AMO communities, as well as of general interest for Nature readers. I would therefore recommend publication of this article, and commend the authors on their impressive results.

We thank the referee again for their positive evaluation of our manuscript and helpful comments.

Additional changes

We have made other very minor changes, separate from the comments of the referees:

1. We have added grant no. W911NF-20-1-0082 to the funding acknowledgement of the Army Research Office MURI.
2. In the Methods section “Movement effects on atom heating and loss”, $\frac{1}{2} \left(1 - \operatorname{erf} \left[(N_{\max} - N) / \sqrt{2N} \right] \right)$ has been changed to $\frac{1}{2} \left(1 + \operatorname{erf} \left[(N_{\max} - N) / \sqrt{2N} \right] \right)$ (since we are technically plotting retention = 1 - loss) and we have made the same change in the Figure legend of ED Fig. 2a,b.

To help offset words added to address the referee’s comments, we have removed words from the following places in the main text:

3. ~~These observations are in excellent agreement with results of exact numerical simulations of the quantum dynamics in the isolated system (lines plotted in Figs. 4c,e and Extended Data Fig. 10).~~

4. ~~We use this method to probe the growth of entanglement entropy produced by many-body dynamics (see Methods for additional benchmarking of the technique, including data on small systems).~~

5. ~~We find that the rate of entropy growth for this initial state is significantly suppressed, and the mutual information reveals an area-law scaling (in contrast to the volume law of the $|ggg\dots\rangle$ quench) (Fig. 4d).~~

6. ~~As an example, Fig. 2a demonstrates preparation of a 1D cluster state, a graph state defined by a linear chain of qubits with edges between neighbors.~~

7. ~~Measuring twins in the Bell basis detects occurrences of the antisymmetric singlet state $|\Psi^-\rangle = \frac{|01\rangle - |10\rangle}{\sqrt{2}}$, whose presence indicates that subsystems of the two copies were in different states due to entanglement with the rest of the many-body system (and entanglement with the environment).~~

8. ~~While such thermalizing dynamics is generically expected in strongly interacting many-body systems, remarkably, this is not always the case. In particular, it was demonstrated previously that for certain initial states this system can evade thermalization. for relatively long times.~~

9. ~~Figure 4 reports the measurement of entanglement properties of many-body scars following a rapid quench from the initial state $|Z_2\rangle \equiv |rgrg\dots\rangle$, initialized by applying local light shifts within one sublattice and performing a global Rydberg π pulses (Methods).~~

10. ~~Furthermore, Fig. 4e shows the single-site entropy in the middle of the chain, demonstrating rapid growth and saturation of entropy for the thermalizing $|ggg\dots\rangle$ state but large oscillations in entropy for the $|Z_2\rangle$ state.~~

11. ~~To realize this state, we perform one global, parallel layer of~~

CZ gates on adjacent atom pairs, move half the atoms to form new pairs, and then perform another parallel layer of CZ gates (Figs. 2a pictures and Fig. 2,b circuit)

12. However, so far progress has been limited to small-scale, few-qubit systems lacking either full connectivity, programmability or true parallelism.

13. Taken together, these ingredients enable a powerful quantum information architecture, which we employ to realize applications including entangled state generation, creation of topological surface and toric code states, and a hybrid analog-digital approach for quantum simulations.

14. Entanglement transport could also find use in empower metrological applications such as creating distributed states for probing gravitational gradients. Finally, our approach can help facilitate quantum networking between separated arrays, paving the way toward large-scale quantum information systems and distributed quantum metrology.

Reviewer Reports on the First Revision:

Referees' comments:

Referee #1 (Remarks to the Author):

The referee thanks the authors for their careful and detailed explanation for all the points of criticisms in the first round, and the referee is happy with all the changes except for point (4), which is still not clarified, and I would like to elaborate on this point in detail:

The referee is fully aware of all the aspects pointed out in the reply. However, their validity requires a very crucial point: the ancilla qubits have to be measured while the coherence of the qubits of the surface code/toric codes is maintained. After this measurement, the topological state is prepared and in each shot a different state is prepared. However, the referee agrees that this state has all the properties of the toric code state and can be used to encode logical qubits.

However reading the manuscript, one gets the impression that all qubits are measured at the same time. This is clearly visible from all the figures, and nowhere in the manuscript this very important aspect is explained. Especially, one would expect that the ability to perform independent measurements and even provide feedback is a highly non-trivial task, and if the current platform is able to achieve this goal it would be another milestone.

Therefore, if the authors really first measure the ancilla qubits, while maintaining the coherence and only later measure the remaining qubits to probe the state, this very important aspect has to be explained in detail. It would be an extremely impressive feature of this Rydberg platform. Especially, the time scale for the measurement, the remaining holding time of the created state should be clearly discussed, and even the dependence of the fidelity of the logical qubit on the holding time should be provided.

In turn, if the authors perform the measurement on all qubits at the same time -- as the referee though from reading the manuscript, the claim that the authors prepare a surface code/toric code state is actually not correct. Then, the authors just prepare a highly entangled cluster state, which can be used in a measurement based approach to create the surface code/toric code.

I think it is very important that this aspect gets clarified.

Referee #2 (Remarks to the Author):

I am content with the corrections made by the authors. I am convinced that this is a milestone paper and recommend accepting it to Nature.

Referee #3 (Remarks to the Author):

I would like to thank the authors for their considered responses to my comments and questions. I am happy with the changes that they made and for publication to proceed.

Referee responses and summary of revisions 2

Nature Manuscript number: 2021-12-19162 Bluvstein

February 7, 2022

Reviewer: 1

The referee thanks the authors for their careful and detailed explanation for all the points of criticisms in the first round, and the referee is happy with all the changes except for point (4), which is still not clarified, and I would like to elaborate on this point in detail:

The referee is fully aware of all the aspects pointed out in the reply. However, their validity requires a very crucial point: the ancilla qubits have to be measured while the coherence of the qubits of the surface code/toric codes is maintained. After this measurement, the topological state is prepared and in each shot a different state is prepared. However, the referee agrees that this state has all the properties of the toric code state and can be used to encode logical qubits.

We thank the referee for their continued thorough questions and comments. In our previous response we explained that while the measured ancilla values are important, they can all be handled in-software for any practical QEC operation and so we indeed prepare the topological toric code state with our method, without any postselection. We are happy that the referee agrees with this point and agrees that no feedforward correction is required to create the logical state; i.e., simply projecting the ancillary qubits and noting their value is sufficient.

However reading the manuscript, one gets the impression that all qubits are measured at the same time. This is clearly visible from all the figures, and nowhere in the manuscript this very important aspect is explained. Especially, one would expect that the ability to perform independent measurements and even provide feedback is a highly non-trivial task, and if the current platform is able to achieve this goal it would be another milestone.

The referee highlights an interesting concept about parallel vs sequential measurements and he/she is correct that in this work we choose to measure all qubits simultaneously (as clearly stated in the first paragraph of the Topological codes section). However, since, as discussed above, no feedforward on data qubits conditional on the ancilla measurements are required for performing quantum operations on encoded states, the order of measurement cannot have any mea-

surable consequences. In other words, the measurement of the ancilla qubits commutes with all data qubit operations following their entanglement with the ancilla qubits. “Projecting” the ancilla qubits before the data qubits cannot affect the state of the data qubits. Therefore the ancilla measurements in the quantum circuit model e.g. in Fig. 3a can be freely slid to any time point following the final $Y(\pi/2)$ pulse (before, or even *after* the data qubit measurements) and the circuit will remain equivalent. We measure all qubits simultaneously for simplicity, as mathematically there cannot be any measurable consequence from the measurement order.

Therefore, if the authors really first measure the ancilla qubits, while maintaining the coherence and only later measure the remaining qubits to probe the state, this very important aspect has to be explained in detail. It would be an extremely impressive feature of this Rydberg platform. Especially, the time scale for the measurement, the remaining holding time of the created state should be clearly discussed, and even the dependence of the fidelity of the logical qubit on the holding time should be provided.

We agree with the referee that in order for feedforward operations to be useful, the ancilla measurement should be done on a timescale much shorter than the lifetime of the data qubits. This is possible in our system, since we can detect the atom state with high-fidelity in ~ 5 ms, which is $300\times$ shorter than the data qubit memory time (1.5 seconds). While such features are very important for certain future applications (such as magic state distillation, see outlook), since no feedforward is required for our method of preparation or use of an error-corrected state, one can measure these state-preparation ancillae at *any* point in time without measurable consequences.

Since this mathematically rigorous statement is indeed somewhat counter-intuitive, to illustrate that our logical qubits are fully functional *before* the ancilla qubits are measured, following the suggestion of the referee we now show the ability of the error correction / detection code to mitigate errors induced during a hold time of the logical state. Specifically, we have added new experimental data (Extended Data Fig. 7b), where we hold on to a created surface code $|+\rangle_L$ state for a variable time. The correction / detection properties of the code are still fully functional, especially visible in the error detection curve which greatly increases the logical qubit lifetime, even though the ancilla qubits are only measured *after* this hold time. This measurement provides direct experimental evidence that we can detect / correct errors that occur after preparing $|+\rangle_L$, *before* the ancillae are measured.

In turn, if the authors perform the measurement on all qubits at the same time – as the referee though from reading the manuscript, the claim that the authors prepare a surface code/toric code state is actually not correct. Then, the authors just prepare a highly entangled cluster state, which can be used in a measurement based approach to create the surface code/toric code.

I think it is very important that this aspect gets clarified.

We thank the Referee for their detailed insights. We hope that the above discussion and the new experimental data (Extended Data Fig. 7) clarify this important issue. To address the referee's comment, we have made the following revisions to the main text:

[Page 7] For these codes the measured values of the ancilla qubits simply redefine the stabilizers and are handled in-software for practical QEC operation [Fowler2012]. Since the redefinition is applied in-software, without physical intervention, the projective measurements on the ancillae commute with all operations on the data qubits and can be done at any time, hence we measure all qubits simultaneously.

We have also added the new sub-figure (Extended Data Fig. 7b) demonstrating error correction / detection on logical dephasing errors introduced during a hold time of the logical qubit. We have added the following reference to the figure, following the surface code data in the main text:

[Page 8] ... (see also Extended Data Fig. 7, showing the expected attributes for all prepared error-protected logical states.)

And to answer the referee's comments about the near-term possibility of mid-circuit readout as a technical tool, in addition to our existing discussion we have added:

[Page 12] Mid-circuit readout can be implemented by moving ancillas into a separate zone and imaging using e.g. avalanche photodiode arrays within a few hundred microseconds [Shea2020] ... We estimate an entire QEC round can be implemented within a millisecond, much faster than the measured $T_2 > 1$ s, and with projected fidelity improvements theoretically surpassing the surface code threshold (Methods). We emphasize that such a mid-circuit readout is essential for realizing scalable fault-tolerant quantum computation.

We believe these revisions address the referee's comments and we thank them again for their positive evaluation of our manuscript, as well as for the continued thoughtfulness of their comments.

Reviewer: 2

I am content with the corrections made by the authors. I am convinced that this is a milestone paper and recommend accepting it to Nature.

We thank the referee again for their positive evaluation of our manuscript.

Reviewer: 3

I would like to thank the authors for their considered responses to my comments and questions. I am happy with the changes that they made and for publication to proceed.

We thank the referee again for their positive evaluation of our manuscript.